# The Best of Both Worlds: Accurate Global and Personalized Models through Federated Learning with Data-Free Hyper-Knowledge Distillation

**Huancheng Chen[1], Chianing Wang[2], Haris Vikalo[1]**
[1]The University of Texas at Austin, TX
[2]Toyota Motor North America, CA

## Abstract

Heterogeneity of data distributed across clients limits the performance of global models trained through federated learning, especially in the settings with highly imbalanced class distributions of local datasets. In recent years, personalized federated learning (pFL) has emerged as a potential solution to the challenges presented by heterogeneous data. However, existing pFL methods typically enhance performance of local models at the expense of the global model's accuracy. We propose FedHKD (Federated Hyper-Knowledge Distillation), a novel FL algorithm in which clients rely on knowledge distillation (KD) to train local models. In particular, each client extracts and sends to the server the means of local data representations and the corresponding soft predictions – information that we refer to as "hyper-knowledge". The server aggregates this information and broadcasts it to the clients in support of local training. Notably, unlike other KD-based pFL methods, FedHKD does not rely on a public dataset nor it deploys a generative model at the server. We analyze convergence of FedHKD and conduct extensive experiments on visual datasets in a variety of scenarios, demonstrating that FedHKD provides significant improvement in both personalized as well as global model performance compared to state-of-the-art FL methods designed for heterogeneous data settings.

## 1 Introduction

Federated learning (FL), a communication-efficient and privacy-preserving alternative to training on centrally aggregated data, relies on collaboration between clients who own local data to train a global machine learning model. A central server coordinates the training without violating clients' privacy – the server has no access to the clients' local data. The first ever such scheme, *Federated Averaging* (FedAvg) (McMahan et al., 2017), alternates between two steps: (1) randomly selected client devices initialize their local models with the global model received from the server, and proceed to train on local data; (2) the server collects local model updates and aggregates them via weighted averaging to form a new global model. As analytically shown in (McMahan et al., 2017), FedAvg is guaranteed to converge when the client data is independent and identically distributed (iid).

A major problem in FL systems emerges when the clients' data is heterogeneous (Kairouz et al., 2021). This is a common setting in practice since the data owned by clients participating in federated learning is likely to have originated from different distributions. In such settings, the FL procedure may converge slowly and the resulting global model may perform poorly on the local data of an individual client. To address this challenge, a number of FL methods aiming to enable learning on non-iid data has recently been proposed (Karimireddy et al., 2020; Li et al., 2020; 2021a; Acar et al., 2021; Liu et al., 2021; Yoon et al., 2021; Chen & Vikalo, 2022). Unfortunately, these methods struggle to train a global model that performs well when the clients' data distributions differ significantly.

Difficulties of learning on non-iid data, as well as the heterogeneity of the clients' resources (e.g., compute, communication, memory, power), motivated a variety of personalized FL (pFL) techniques

(Arivazhagan et al., 2019; T Dinh et al., 2020; Zhang et al., 2020; Huang et al., 2021; Collins et al., 2021; Tan et al., 2022). In a pFL system, each client leverages information received from the server and utilizes a customized objective to locally train its personalized model. Instead of focusing on global performance, a pFL client is concerned with improving the model's local performance empirically evaluated by running the local model on data having distribution similar to the distribution of local training data. Since most personalized FL schemes remain reliant upon on gradient or model aggregation, they are highly susceptible to 'stragglers' that slow down the training convergence process. FedProto (Tan et al., 2021) is proposed to address high communication cost and limitations of homogeneous models in federated learning. Instead of model parameters, in FedProto each client sends to the server only the class prototypes – the means of the representations of the samples in each class. Aggregating the prototypes rather than model updates significantly reduces communication costs and lifts the requirement of FedAvg that clients must deploy the same model architecture. However, note that even though FedProto improves local validation accuracy by utilizing aggregated class prototypes, it leads to barely any improvement in the global performance. Motivated by the success of Knowledge Distillation (KD) (Hinton et al., 2015) which infers soft predictions of samples as the 'knowledge' extracted from a neural network, a number of FL methods that aim to improve global model's generalization ability has been proposed (Jeong et al., 2018b; Li & Wang, 2019; Lin et al., 2020; Zhang et al., 2021). However, most of the existing KD-based FL methods require that a public dataset is provided to all clients, limiting the feasibility of these methods in practical settings.

In this paper we propose FedHKD (Federated Hyper-Knowledge Distillation), a novel FL framework that relies on prototype learning and knowledge distillation to facilitate training on heterogeneous data. Specifically, the clients in FedHKD compute mean representations and the corresponding mean soft predictions for the data classes in their local training sets; this information, which we refer to as "hyper-knowledge," is endued by differential privacy via the Gaussian mechanism and sent for aggregation to the server. The resulting globally aggregated hyper-knowledge is used by clients in the subsequent training epoch and helps lead to better personalized and global performance. A number of experiments on classification tasks involving SVHN (Netzer et al., 2011), CIFAR10 and CIFAR100 datasets demonstrate that FedHKD consistently outperforms state-of-the-art approaches in terms of both local and global accuracy.

## 2 RELATED WORK

### 2.1 HETEROGENEOUS FEDERATED LEARNING

Majority of the existing work on federated learning across data-heterogeneous clients can be organized in three categories. The first set of such methods aims to reduce variance of local training by introducing regularization terms in local objective (Karimireddy et al., 2020; Li et al., 2020; 2021a; Acar et al., 2021). (Mendieta et al., 2022) analyze regularization-based FL algorithms and, motivated by the regularization technique GradAug in centralized learning (Yang et al., 2020), propose FedAlign. Another set of techniques for FL on heterogeneous client data aims to replace the naive model update averaging strategy of FedAvg by more efficient aggregation schemes. To this end, PFNM (Yurochkin et al., 2019) applies a Bayesian non-parametric method to select and merge multi-layer perceptron (MLP) layers from local models into a more expressive global model in a layer-wise manner. FedMA ((Wang et al., 2020a)) proceeds further in this direction and extends the same principle to CNNs and LSTMs. (Wang et al., 2020b) analyze convergence of heterogeneous federated learning and propose a novel normalized averaging method. Finally, the third set of methods utilize either the mixup mechanism (Zhang et al., 2017) or generative models to enrich diversity of local datasets (Yoon et al., 2021; Liu et al., 2021; Chen & Vikalo, 2022). However, these methods introduce additional memory/computation costs and increase the required communication resources.

### 2.2 PERSONALIZED FEDERATED LEARNING

Motivated by the observation that a global model collaboratively trained on highly heterogeneous data may not generalize well on clients' local data, a number of personalized federated learning (pFL) techniques aiming to train customized local models have been proposed (Tan et al., 2022). They can be categorized into two groups depending on whether or not they also train a global model. The pFL techniques focused on global model personalization follow a procedure similar to the plain vanilla FL – clients still need to upload all or a subset of model parameters to the server to enable global model aggregation. The global model is personalized by each client via local adaptation

steps such as fine-tuning (Wang et al., 2019; Hanzely et al., 2020; Schneider & Vlachos, 2021), creating a mixture of global and local layers (Arivazhagan et al., 2019; Mansour et al., 2020; Deng et al., 2020; Zec et al., 2020; Hanzely & Richtárik, 2020; Collins et al., 2021; Chen & Chao, 2021), regularization (T Dinh et al., 2020; Li et al., 2021b) and meta learning (Jiang et al., 2019; Fallah et al., 2020). However, when the resources available to different clients vary, it is impractical to require that all clients train models of the same size and type. To address this, some works waive the global model by adopting multi-task learning (Smith et al., 2017) or hyper-network frameworks (Shamsian et al., 2021). Inspired by prototype learning (Snell et al., 2017; Hoang et al., 2020; Michieli & Ozay, 2021), FedProto (Tan et al., 2021) utilizes aggregated class prototypes received from the server to align clients' local objectives via a regularization term; since there is no transmission of model parameters between clients and the server, this scheme requires relatively low communication resources. Although FedProto improves local test accuracy of the personalized models, it does not benefit the global performance.

### 2.3 Federated learning with Knowledge Distillation

Knowledge Distillation (KD) (Hinton et al., 2015), a technique capable of extracting knowledge from a neural network by exchanging soft predictions instead of the entire model, has been introduced to federated learning to aid with the issues that arise due to variations in resources (computation, communication and memory) available to the clients (Jeong et al., 2018a; Chang et al., 2019; Itahara et al., 2020). FedMD (Li & Wang, 2019), FedDF (Lin et al., 2020) and FedKT-pFL (Zhang et al., 2021) transmit only soft-predictions as the knowledge between the server and clients, allowing for personalized/heterogeneous client models. However, these KD-based federated learning methods require that a public dataset is made available to all clients, presenting potential practical challenges. Recent studies (Zhu et al., 2021; Zhang et al., 2022) explored using GANs (Goodfellow et al., 2014) to enable data-free federated knowledge distillation in the context of image classification tasks; however, training GANs incurs considerable additional computation and memory requirements.

In summary, most of the existing KD-based schemes require a shared dataset to help align local models; others require costly computational efforts to synthesize artificial data or deploy a student model at the server and update it using local gradients computed when minimizing the divergence of soft prediction on local data between clients' teacher model and the student model (Lin et al., 2020). In our framework, we extend the concept of knowledge to 'hyper-knowledge', combining class prototypes and soft predictions on local data to improve both the local test accuracy and global generalization ability of federated learning.

## 3 Methodology

### 3.1 Problem Formulation

Consider a federated learning system where $m$ clients own local private dataset $\mathcal{D}_1, \ldots, \mathcal{D}_m$; the distributions of the datasets may vary across clients, including the scenario in which a local dataset contains samples from only a fraction of classes. In such an FL system, the clients communicate locally trained models to the server which, in turn, sends the aggregated global model back to the clients. The plain vanilla federated learning (McMahan et al., 2017) implements aggregation as

$$w^t = \sum_{i=1}^{m} \frac{|\mathcal{D}_i|}{M} w_i^{t-1}, \tag{1}$$

where $w^t$ denotes parameters of the global model at round $t$; $w_i^{t-1}$ denotes parameters of the local model of client $i$ at round $t-1$; $m$ is the number of participating clients; and $M = \sum_{i=1}^{m} |\mathcal{D}_i|$. The clients are typically assumed to share the same model architecture. Our aim is to learn a personalized model $w_i$ for each client $i$ which not only performs well on data generated from the distribution of the $i^{th}$ client's local training data, but can further be aggregated into a global model $w$ that performs well across all data classes (i.e., enable accurate global model performance). This is especially difficult when the data is heterogenous since straightforward aggregation in such scenarios likely leads to inadequate performance of the global model.

### 3.2 Utilizing Hyper-Knowledge

Knowledge distillation (KD) based federated learning methods that rely on a public dataset require clients to deploy local models to run inference / make predictions for the samples in the public

dataset; the models' outputs are then used to form soft predictions according to

$$q_i = \frac{\exp(z_i/T)}{\sum_j \exp(z_j/T)}, \tag{2}$$

where $z_i$ denotes the $i^{\text{th}}$ element in the model's output $\boldsymbol{z}$ for a given data sample; $q_i$ is the $i^{\text{th}}$ element in the soft prediction $\boldsymbol{q}$; and $T$ is the so-called "temperature" parameter. The server collects soft predictions from clients (local knowledge), aggregates them into global soft predictions (global knowledge), and sends them to clients to be used in the next training round. Performing inference on the public dataset introduces additional computations in each round of federated learning, while sharing and locally storing public datasets consumes communication and memory resources. It would therefore be beneficial to develop KD-based methods that do not require use of public datasets; synthesizing artificial data is an option, but one that is computationally costly and thus may be impractical. To this end, we extend the notion of distilled knowledge to include both the averaged representations and the corresponding averaged soft predictions, and refer to it as "hyper-knowledge"; the "hyper-knowledge" is protected via the Gaussian differential privacy mechanism and shared between clients and server.

**Feature Extractor and Classifier.** We consider image classification as an illustrative use case. Typically, a deep network for classification tasks consists of two parts (Kang et al., 2019): (1) a feature extractor translating the input raw data (i.e., an image) into latent space representation; (2) a classifier mapping representations into categorical vectors. Formally,

$$\boldsymbol{h}_i = R_{\boldsymbol{\phi}_i}(\boldsymbol{x}_i), \quad \boldsymbol{z}_i = G_{\boldsymbol{\omega}_i}(\boldsymbol{h}_i), \tag{3}$$

where $\boldsymbol{x}_i$ denotes raw data of client $i$, $R_{\boldsymbol{\phi}_i}(\cdot)$ and $G_{\boldsymbol{\omega}_i}(\cdot)$ are the embedding functions of feature extractor and classifier with model parameters $\boldsymbol{\phi}_i$ and $\boldsymbol{\omega}_i$, respectively; $\boldsymbol{h}_i$ is the representation vector of $\boldsymbol{x}_i$; and $\boldsymbol{z}_i$ is the categorical vector.

**Evaluating and Using Hyper-Knowledge.** The mean latent representation of class $j$ in the local dataset of client $i$ is computed as

$$\bar{\boldsymbol{h}}_i^j = \frac{1}{N_i^j} \sum_{k=1}^{N_i^j} \boldsymbol{h}_i^{j,k}, \quad \bar{\boldsymbol{q}}_i^j = \frac{1}{N_i^j} \sum_{k=1}^{N_i^j} Q(\boldsymbol{z}_i^{j,k}, T) \tag{4}$$

where $N_i^j$ is the number of samples with label $j$ in client $i$'s dataset; $Q(\cdot, T)$ is the soft target function; $\boldsymbol{h}_i^{j,k}$ and $\boldsymbol{z}_i^{j,k}$ are the data representation and prediction of the $i^{\text{th}}$ client's $k^{\text{th}}$ sample with label $j$. The mean latent data representation $\bar{\boldsymbol{h}}_i^j$ and soft prediction $\bar{\boldsymbol{q}}_i^j$ are the hyper-knowledge of class $j$ in client $i$; for convenience, we denote $\mathcal{K}_i^j = (\bar{\boldsymbol{h}}_i^j, \bar{\boldsymbol{q}}_i^j)$. If there are $n$ classes, then the full hyper-knowledge of client $i$ is $\mathcal{K}_i = \{\mathcal{K}_i^1, \ldots, \mathcal{K}_i^n\}$. As a comparison, FedProto (Tan et al., 2021) only utilizes means of data representations and makes no use of soft predictions. Note that to avoid the situations where $\mathcal{K}_i^j = \emptyset$, which may happen when data is highly heterogeneous, FedHKD sets a threshold (tunable hyper-parameter) $\nu$ which is used to decided whether or not a client should share its hyper-knowledge; in particular, if the fraction of samples with label $j$ in the local dataset of client $i$ is below $\nu$, client $i$ is not allowed to share the hyper-knowledge $\mathcal{K}_i^j$. If there is no participating client sharing hyper-knowledge for class $j$, the server sets $\mathcal{K}^j = \emptyset$. A flow diagram illustrating the computation of hyper-knowledge is given in Appendix. A.3.

**Differential Privacy Mechanism.** It has previously been argued that communicating averaged data representation promotes privacy (Tan et al., 2021); however, hyper-knowledge exchanged between server and clients may still be exposed to differential attacks (Dwork, 2008; Geyer et al., 2017). A number of studies (Geyer et al., 2017; Sun et al., 2021; Gong et al., 2021; Ribero et al., 2022; Chen & Vikalo, 2022) that utilize differential privacy to address security concerns in federated learning have been proposed. The scheme presented in this paper promotes privacy by protecting the shared means of data representations through a differential privacy (DP) mechanism (Dwork et al., 2006a;b) defined below.

**Definition 1 (($\varepsilon, \delta$)-Differential Privacy)** *A randomized function $\mathcal{F} : \mathcal{D} \to \mathbb{R}$ provides ($\varepsilon, \delta$)-differential privacy if for all adjacent datasets $\boldsymbol{d}, \boldsymbol{d}' \in \mathcal{D}$ differing on at most one element, and all $\boldsymbol{S} \in range(\mathcal{F})$, it holds that*

$$\mathbb{P}[\mathcal{F}(\boldsymbol{d}) \in \boldsymbol{S}] \leq e^{\epsilon} \mathbb{P}[\mathcal{F}(\boldsymbol{d}') \in \boldsymbol{S}] + \delta, \tag{5}$$

*where $\epsilon$ denotes the maximum distance between the range of $\mathcal{F}(d)$ and $\mathcal{F}(d')$ and may be thought of as the allotted privacy budget, while $\delta$ is the probability that the maximum distance is not bounded by $\varepsilon$.*

Any deterministic function $f : \mathcal{D} \to \mathbb{R}$ can be endued with arbitrary $(\epsilon, \delta)$-differential privacy via the Gaussian mechanism, defined next.

**Theorem 1 (Gaussian mechanism)** *A randomized function $\mathcal{F}$ derived from any deterministic function $f : \mathcal{D} \to \mathbb{R}$ perturbed by Gaussian noise $\mathcal{N}(0, S_f^2 \cdot \sigma^2)$,*

$$\mathcal{F}(\boldsymbol{d}) = f(\boldsymbol{d}) + \mathcal{N}\left(0, S_f^2 \cdot \sigma^2\right), \tag{6}$$

*achieves $(\varepsilon, \delta)$-differential privacy for any $\sigma > \sqrt{2 \log \frac{5}{4\delta}}/\varepsilon$. Here $S_f$ denotes the sensitivity of function $f$ defined as the maximum of the absolute distance $|f(\boldsymbol{d}) - f(\boldsymbol{d}')|$.*

We proceed by defining a deterministic function $f_l(\boldsymbol{d}_i^j) \triangleq \bar{\boldsymbol{h}}_i^j(l) = \frac{1}{N_i^j} \sum_{k=1}^{N_i^j} \boldsymbol{h}_i^{j,k}(l)$ which evaluates the $l^{\text{th}}$ element of $\bar{\boldsymbol{h}}_i^j$, where $\boldsymbol{d}_i^j$ is the subset of client $i$'s local dataset including samples with label $j$ only; $\boldsymbol{h}_i^{j,k}$ denotes the representation of the $k^{\text{th}}$ sample in $\boldsymbol{d}_i^j$ while $\boldsymbol{h}_i^{j,k}(l)$ is the $l^{\text{th}}$ element of $\boldsymbol{h}_i^{j,k}$. In our proposed framework, client $i$ transmits noisy version of its hyper-knowledge to the server,

$$\tilde{\boldsymbol{h}}_i^j(l) = \bar{\boldsymbol{h}}_i^j(l) + \boldsymbol{\chi}_i^j(l), \tag{7}$$

where $\boldsymbol{\chi}_i^j(l) \sim \mathcal{N}(0, (S_f^i)^2 \cdot \sigma^2)$; $\sigma^2$ denotes a hyper-parameter shared by all clients. $(S_f^i)^2$ is the sensitive of function $f_l(\cdot)$ with client $i$'s local dataset.

**Lemma 1** *If $|\boldsymbol{h}_i^{j,k}(l)|$ is bounded by $\zeta > 0$ for any $k$, then*

$$|f_l(\boldsymbol{d}_i^j) - f_l(\boldsymbol{d}_i^{j'})| \leq \frac{2\zeta}{N_i^j} \tag{8}$$

Therefore, $S_f^i = \frac{2\zeta}{N_i^j}$. Note that $(S_f^i)^2$ depends on $N_i^j$, the number of samples in class $j$, and thus differs across clients in the heterogeneous setting. A discussion on the probability that differential privacy is broken can be found in the Section 4.3. Proof of Lemma 1 is provided in Appendix A.5.

## 3.3 GLOBAL HYPER-KNOWLEDGE AGGREGATION

After the server collects hyper-knowledge from participating clients, the global hyper-knowledge for class $j$ at global round $t + 1$, $\mathcal{K}^{j,t+1} = \left(\mathcal{H}^{j,t+1}, \mathcal{Q}^{j,t+1}\right)$, is formed as

$$\mathcal{H}^{j,t+1} = \sum_{i=1}^{m} p_i \tilde{\boldsymbol{h}}_i^{j,t}, \quad \mathcal{Q}^{j,t+1} = \sum_{i=1}^{m} p_i \bar{\boldsymbol{q}}_i^{j,t}, \tag{9}$$

where $p_i = N_i^j/N^j$, $N_i^j$ denotes the number of samples in class $j$ owned by client $i$, and $N^j = \sum_{i=1}^{m} N_i^j$. For clarity, we emphasize that $\tilde{\boldsymbol{h}}_i^{j,t}$ denotes the local hyper-knowledge about class $j$ of client $i$ at global round $t$. Since the noise is drawn from $\mathcal{N}\left(0, (S_f^i)^2 \cdot \sigma^2\right)$, its effect on the quality of hyper-knowledge is alleviated during aggregation assuming sufficiently large number of participating clients, i.e.,

$$\mathbb{E}\left[\mathcal{H}^{j,t+1}(l)\right] = \sum_{i=1}^{m} p_i \bar{\boldsymbol{h}}_i^{j,t}(l) + \mathbb{E}\left[\sum_{i=1}^{m} p_i \boldsymbol{\chi}_i^{j,t}(l)\right] = \sum_{i=1}^{m} p_i \bar{\boldsymbol{h}}_i^{j,t}(l) + 0, \tag{10}$$

with variance $\sigma^2 \sum_{i=1}^{m}(S_f^i)^2$. In other words, the additive noise is "averaged out" and effectively near-eliminated after aggregating local hyper-knowledge. For simplicity, we assume that in the above expressions $N_i^j \neq 0$.

## 3.4 LOCAL TRAINING OBJECTIVE

Following the aggregation at the server, the global hyper-knowledge is sent to the clients participating in the next FL round to assist in local training. In particular, given data samples $(\boldsymbol{x}, y) \sim \mathcal{D}_i$, the loss function of client $i$ is formed as

$$\mathcal{L}(\mathcal{D}_i, \boldsymbol{\phi}_i, \boldsymbol{\omega}_i) = \frac{1}{B_i} \sum_{k=1}^{B_i} \mathbf{CELoss}(G_{\boldsymbol{\omega}_i}(R_{\boldsymbol{\phi}_i}(\boldsymbol{x}_k)), y_k)$$

$$+ \lambda \frac{1}{n} \sum_{j=1}^{n} ||Q(G_{\boldsymbol{\omega}_i}(\mathcal{H}^j), T) - \mathcal{Q}^j||_2 + \gamma \frac{1}{B_i} \sum_{k=1}^{B_i} ||R_{\boldsymbol{\phi}_i}(\boldsymbol{x}_k) - \mathcal{H}^{y_k}||_2 \tag{11}$$

where $B_i$ denotes the number of samples in the dataset owned by client $i$, $n$ is the number of classes, **CELoss**$(\cdot, \cdot)$ denotes the cross-entropy loss function, $\| \cdot \|_2$ denotes Euclidean norm, $Q(\cdot, T)$ is the soft target function with temperature $T$, and $\lambda$ and $\gamma$ are hyper-parameters.

Note that the loss function in (11) consists of three terms: the empirical risk formed using predictions and ground-truth labels, and two regularization terms utilizing hyper-knowledge. Essentially, the second and third terms in the loss function are proximity/distance functions. The second term is to force the local classifier to output similar soft predictions when given global data representations while the third term is to force the features extractor to output similar data representations when given local data samples. For both, we use Euclidean distance because it is non-negative and convex.

### 3.5 FedHKD: Summary of the Framework

The training starts at the server by initializing the global model $\boldsymbol{\theta}^1 = (\boldsymbol{\phi}^1, \boldsymbol{\omega}^1)$, where $\boldsymbol{\phi}^1$ and $\boldsymbol{\omega}^1$ denote parameters of the global feature extractor and global classifier, respectively. At the beginning of each global epoch, the server sends the global model and global hyper-knowledge to clients selected for training. In turn, each client initializes its local model with the received global model, and performs updates by minimizing the objective in Eq. 11; the objective consists of three terms: (1) prediction loss in a form of the cross-entropy between prediction and ground-truth; (2) classifier loss reflective of the Euclidean norm distance between the output of the classifier and the corresponding global soft predictions; and (3) feature loss given by the Euclidean norm distance between representations extracted from raw data by a local feature extractor and global data representations. Having completed local updates, clients complement their local hyper-knowledge by performing inference on local data, and finally send local model as well as local hyper-knowledge to the server for aggregation. The method outlined in this section is formalized as Algorithm 1. For convenience, we provided a visualization of the FedHKD procedure in Appendix. A.4.

---

**Algorithm 1** FedHKD

---

**Input:**
  Datasets distributed across $m$ clients, $\mathcal{D} = \{\mathcal{D}_1, \mathcal{D}_2, \ldots \mathcal{D}_m\}$; client participating rate $\mu$; hyper-parameters $\lambda$ and $\gamma$; the sharing threshold $\nu$; variance $\sigma^2$ characterizing differential privacy noise; temperature $T$; the number of global epochs $T_r$.
**Output:**
  The global model $\boldsymbol{\theta}^{T_r+1} = (\boldsymbol{\phi}^{T_r+1}, \boldsymbol{\omega}^{T_r+1})$

1: **Server executes:**
2: randomly initialize $(\boldsymbol{\phi}^1, \boldsymbol{\omega}^1)$, $\mathcal{K} = \{\}$
3: **for** $t = 1, \ldots, T_r$ **do**
4:   $\mathcal{S}_t \leftarrow \lfloor m\mu \rfloor$ clients selected at random
5:   send the global model $\boldsymbol{\phi}^t, \boldsymbol{\omega}^t, \mathcal{K}$ to clients in $\mathcal{S}_t$
6:   **for** $i \in \mathcal{S}_t$ **do**
7:     $\boldsymbol{\phi}_i^t, \boldsymbol{\omega}_i^t, \mathcal{K}_i \leftarrow$ **LocalUpdate**$(\boldsymbol{\phi}^t, \boldsymbol{\omega}^t, \mathcal{K}, \mathcal{D}_i, \sigma^2, \nu, i)$
8:   **end for**
9:   Aggregate global hyper-knowledge $\mathcal{K}$ by Eq. 9.
10:   Aggregate global model $\boldsymbol{\theta}^{t+1} = (\boldsymbol{\phi}^{t+1}, \boldsymbol{\omega}^{t+1})$
11: **end for**
12: **return** $\boldsymbol{\theta}^{T_r+1} = (\boldsymbol{\phi}^{T_r+1}, \boldsymbol{\omega}^{T_r+1})$
13:
14: **LocalUpdate**$(\boldsymbol{\phi}^t, \boldsymbol{\omega}^t, \mathcal{K}, \mathcal{D}_i, \sigma_s^2, i)$:
15: $\boldsymbol{\phi}_i^t \leftarrow \boldsymbol{\phi}^t, \boldsymbol{\omega}_i^t \leftarrow \boldsymbol{\omega}^t, (x, y) \sim \mathcal{D}_i$
16: **for** each local epoch **do**
17:   $\boldsymbol{\phi}_i^t, \boldsymbol{\omega}_i^t \leftarrow$ **OptimAlg**$(\mathcal{L}(x, y, \mathcal{K}, \lambda, \gamma))$
18: **end for**
19: update local hyper-knowledge $\mathcal{K}_i$
20: **return** $\boldsymbol{\phi}_i^t, \boldsymbol{\omega}_i^t, \mathcal{K}_i$

---

### 3.6 Convergence Analysis

To facilitate the convergence analysis of FedHKD, we make the assumptions commonly encountered in literature (Li et al., 2019; 2020; Tan et al., 2021). The details in assumptions and proof are in Appendix A.6.

**Theorem 2.** Instate Assumptions 1-3 A.6.1. For an arbitrary client, after each communication round the loss function is bounded as

$$\mathbb{E}\left[\mathcal{L}_i^{\frac{1}{2}, t+1}\right] \leq \mathcal{L}_i^{\frac{1}{2}, t} - \sum_{e=\frac{1}{2}}^{E-1} \left(\eta_e - \frac{\eta_e^2 L_1}{2}\right) \left\|\nabla \mathcal{L}^{e,t}\right\|_2^2 + \frac{\eta_0^2 L_1 E}{2} \left(EV^2 + \sigma^2\right)$$
$$+ 2\lambda\eta_0 L_3 (L_2 + 1) EV + 2\gamma\eta_0 L_2 EV. \tag{12}$$

**Theorem 3.** (FedHKD convergence rate) Instate Assumptions 1-3 A.6.1 hold and define regret $\Delta = \mathcal{L}^{\frac{1}{2},1} - \mathcal{L}^*$. If the learning rate is set to $\eta$, for an arbitrary client after

$$T = \frac{2\Delta}{\epsilon E \left(2\eta - \eta^2 L_1\right) - \eta^2 L_1 E \left(EV^2 + \sigma^2\right) - 4\lambda\eta L_3 \left(L_2 + 1\right) EV - 4\gamma\eta L_2 EV} \tag{13}$$

global rounds ($\epsilon > 0$), it holds that

$$\frac{1}{TE} \sum_{t=1}^{T} \sum_{e=\frac{1}{2}}^{E-1} \left\| \nabla\mathcal{L}^{e,t} \right\|_2^2 \le \epsilon, \tag{14}$$

## 4 EXPERIMENTS

### 4.1 EXPERIMENTAL SETTINGS

In this section, we present extensive benchmarking results comparing the performance of FedHKD and the competing FL methods designed to address the challenge of learning from non-iid data. All the methods were implemented and simulated in Pytorch (Paszke et al., 2019), with models trained using Adam optimizer (Kingma & Ba, 2014). Details of the implementation and the selection of hyper-parameters are provided in Appendix. Below we describe the datasets, models and baselines used in the experiments.

**Datasets.** Three benchmark datasets are used in the experiments: SVHN (Netzer et al., 2011), CIFAR10 and CIFAR100 (Krizhevsky et al., 2009). To generate heterogeneous partitions of local training data, we follow the strategy in (Yoon et al., 2021; Yurochkin et al., 2019; Li et al., 2021a) and utilize Dirichlet distribution with varied concentration parameters $\beta$ which controls the level of heterogeneity. Since our focus is on understanding and addressing the impact of class heterogeneity in clients data on the performance of trained models, we set equal the size of clients' datasets. Furthermore, to evaluate both personalized as well as global model performance, each client is allocated a local test dataset (with the same class distribution as the corresponding local training dataset) and a global test dataset with uniformly distributed classes (shared by all participating clients); this allows computing both the average local test accuracy of the trained local models as well as the global test accuracy of the global model aggregated from the clients' local models.

**Models.** Rather than evaluate the performance of competing schemes on a simple CNN network as in (McMahan et al., 2017; Li et al., 2020; 2021a), we apply two widely used benchmarking models better suited to practical settings. Specifically, we deploy ShuffleNetV2 (Ma et al., 2018) on SVHN and ResNet18 (He et al., 2016) on CIFAR10/100. As our results show, FedHKD generally outperforms competing methods on both (very different) architectures, demonstrating remarkable consistency and robustness.

**Baselines.** We compare the test accuracy of FedHKD with seven state-of-the-art federated learning methods including FedAvg (McMahan et al., 2017), FedMD (Li & Wang, 2019), FedProx (Li et al., 2020), Moon (Li et al., 2021a), FedProto (Tan et al., 2021), FedGen (Zhu et al., 2021) and FedAlign (Mendieta et al., 2022). We emphasize that the novelty of FedHKD lies in data-free knowledge distillation that requires neither a public dataset nor a generative model; this stands in contrast to FedMD which relies on a public dataset and FedGen which deploys a generative model. Like FedHKD, FedProto shares means of data representations but uses different regularization terms in the loss functions and does not make use of soft predictions. When discussing the results, we will particularly analyze and compare the performance of FedMD, FedGen and FedProto with the performance of FedHKD.

### 4.2 PERFORMANCE ANALYSIS

Table 1 shows that FedHKD generally outperforms other methods across various settings and datasets. For each dataset, we ran experiments with 10, 20 and 50 clients, with local data generated from a Dirichlet distribution with fixed concentration parameter $\beta = 0.5$. As previously stated, we focus on the heterogeneity in class distribution of local dataset rather than the heterogeneity in the number of samples. To this end, an increasing fraction of data is partitioned and allocated to the clients in the experiments, maintaining the size of local datasets as the number of clients increases. A single client's averaged training time per global round is computed across different settings to characterize the required training time. To provide a more informative comparison with FedProto (Tan

Table 1: Results on data partitions generated from Dirichlet distribution with the concentration parameter $\beta = 0.5$. The number of clients is 10, 20 and 50; the clients utilize 10%, 20% and 50% of the datasets. The number of parameters (in millions) indicates the size of the model stored in the memory during training. A single client's averaged wall-clock time per round is measured across 8 AMD Vega20 GPUs in a parallel manner.

| Dataset | Scheme | Local Acc | | | Global Acc | | | Params (M) | Time (s) | Pub Data |
|---------|--------|-----------|-----|-----|------------|-----|-----|------------|----------|----------|
| | # Clients | 10 | 20 | 50 | 10 | 20 | 50 | | | |
| SVHN | FedAvg | 0.6766 | 0.7329 | 0.6544 | 0.4948 | 0.6364 | 0.5658 | 1.286 | 5.22 | No |
| | FedProx | 0.6927 | 0.6717 | 0.6991 | 0.5191 | 0.6419 | 0.6139 | 2.572 | 5.56 | No |
| | Moon | 0.6602 | 0.7085 | 0.7192 | 0.4883 | 0.5536 | 0.6543 | 3.858 | 12.32 | No |
| | FedAlign | 0.7675 | 0.7920 | 0.7656 | 0.6426 | 0.7138 | 0.7437 | 1.286 | 16.67 | No |
| | FedGen | 0.5788 | 0.5658 | 0.4679 | 0.3622 | 0.3421 | 0.3034 | 1.357 | 6.66 | No |
| | FedMD | 0.8038 | 0.8086 | 0.7912 | **0.6812** | 0.7344 | **0.8085** | 1.286 | 10.67 | **Yes** |
| | FedProto | 0.8071 | 0.8148 | 0.8039 | 0.6064 | 0.6259 | 0.7895 | 1.286 | 5.42 | No |
| | FedHKD* | 0.8064 | 0.8157 | **0.8072** | 0.6405 | 0.6884 | 0.7921 | 1.286 | 5.70 | No |
| | FedHKD | **0.8086** | **0.8381** | 0.7891 | 0.6781 | **0.7357** | 0.7891 | 1.286 | 6.33 | No |
| CIFAR10 | FedAvg | 0.5950 | 0.6261 | 0.5825 | 0.4741 | 0.5516 | 0.3773 | 11.209 | 8.71 | No |
| | FedProx | 0.5981 | 0.6295 | 0.6490 | 0.4793 | 0.5258 | 0.5348 | 22.418 | 10.25 | No |
| | Moon | 0.5901 | 0.6482 | 0.5513 | 0.4579 | 0.5651 | 0.3514 | 33.627 | 20.52 | No |
| | FedAlign | 0.5948 | 0.6023 | 0.6402 | 0.4976 | 0.5134 | 0.5641 | 11.209 | 36.24 | No |
| | FedGen | 0.5879 | 0.6395 | 0.6533 | 0.4800 | 0.5408 | 0.5651 | 11.281 | 10.52 | No |
| | FedMD | 0.6147 | 0.6666 | 0.6533 | 0.5088 | 0.5575 | **0.5714** | 11.209 | 22.51 | **Yes** |
| | FedProto | 0.6131 | 0.6505 | 0.5939 | 0.5012 | 0.5548 | 0.4016 | 11.209 | 11.68 | No |
| | FedHKD* | 0.6227 | 0.6515 | **0.6675** | 0.5049 | 0.5596 | 0.5074 | 11.209 | 11.26 | No |
| | FedHKD | **0.6254** | **0.6816** | 0.6671 | **0.5213** | **0.5735** | 0.5493 | 11.209 | 12.83 | No |
| CIFAR100 | FedAvg | 0.2361 | 0.2625 | 0.2658 | 0.2131 | 0.2748 | 0.2907 | 11.215 | 14.17 | No |
| | FedProx | 0.2332 | 0.2814 | 0.2955 | 0.2267 | 0.2708 | 0.2898 | 22.430 | 19.81 | No |
| | Moon | 0.2353 | 0.2729 | 0.2428 | 0.2141 | 0.2652 | 0.1928 | 33.645 | 36.28 | No |
| | FedAlign | 0.2467 | 0.2617 | 0.2854 | 0.2281 | 0.2729 | 0.2933 | 11.215 | 27.61 | No |
| | FedGen | 0.2393 | 0.2701 | 0.2739 | 0.2176 | 0.262 | 0.2739 | 11.333 | 17.45 | No |
| | FedMD | 0.2681 | 0.3054 | 0.3293 | 0.2323 | 0.2669 | 0.2968 | 11.215 | 29.04 | **Yes** |
| | FedProto | 0.2568 | 0.3188 | 0.3170 | 0.2121 | 0.2756 | 0.2805 | 11.215 | 14.88 | No |
| | FedHKD* | 0.2551 | 0.2997 | 0.3016 | 0.2286 | 0.2715 | 0.2976 | 11.215 | 14.59 | No |
| | FedHKD | **0.2981** | **0.3245** | **0.3375** | **0.2369** | **0.2795** | **0.2988** | 11.215 | 15.14 | No |

Table 2: Results on data partitions generated with different concentration parameters (10 clients).

| Scheme | Local Acc | | Global Acc | | Local Acc | | Global Acc | |
|--------|-----------|-----|------------|-----|-----------|-----|------------|-----|
| | CIFAR10 | | | | SVHN | | | |
| | $\beta = 0.2$ | $\beta = 5$ | $\beta = 0.2$ | $\beta = 5$ | $\beta = 0.2$ | $\beta = 5$ | $\beta = 0.2$ | $\beta = 5$ |
| FedAvg | 0.5917 | 0.4679 | 0.3251 | 0.5483 | 0.6227 | 0.5833 | 0.2581 | 0.6238 |
| FedProx | 0.6268 | 0.4731 | 0.3845 | 0.5521 | 0.7481 | 0.6598 | 0.4323 | 0.7121 |
| Moon | 0.5762 | 0.3794 | 0.3229 | 0.4256 | 0.7440 | 0.6568 | 0.3764 | 0.7128 |
| FedAlign | 0.6434 | 0.4799 | 0.4446 | 0.5526 | 0.8161 | 0.7414 | 0.5904 | 0.7919 |
| FedGen | 0.6212 | 0.4432 | 0.4623 | 0.4432 | 0.7248 | 0.6542 | 0.5304 | 0.7251 |
| FedMD | 0.6532 | 0.494 | 0.4408 | 0.5543 | 0.8415 | **0.7580** | 0.6181 | **0.8144** |
| FedProto | 0.6471 | 0.4802 | 0.3887 | 0.5488 | 0.8446 | 0.7363 | 0.5493 | 0.8055 |
| FedHKD* | 0.6798 | 0.4857 | 0.4459 | 0.5494 | 0.8344 | 0.7314 | 0.5357 | 0.8044 |
| FedHKD | **0.6789** | **0.4976** | **0.4736** | **0.5573** | **0.8462** | 0.7420 | **0.6241** | 0.8083 |

et al., 2021), we ran two setting of our proposed method, labeled as FedHKD and FedHKD*: (1) FedHKD deploys the second and third term in Eq. 11 using $\lambda = 0.05$ and $\gamma = 0.05$; (2) FedHKD* excludes the constraint on Feature Extractor $R_\phi$ by setting $\lambda = 0.05$ and $\gamma = 0$.

**Accuracy comparison.** The proposed method, FedHKD, generally ranks as either the best or the second best in terms of both local and global accuracy, competing with FedMD without using public data. On SVHN, FedHKD significantly improves the local test accuracy over FedAvg (by 19.5%, 14.3% and 20.6%) as well as the global test accuracy (by 37.0%, 15.6% and 39.5%) in experiments involving 10, 20 and 50 clients, respectively. The improvement over FedAvg carry over to the experiments on CIFAR10, with 5.1%, 8.9% and 14.5% increase in local accuracy and 14.5%, 9.9% and 45.6% increase in global accuracy in the experiments involving 10, 20 and 50 clients, respectively. On CIFAR100, the improvement of global accuracy is somewhat more modest, but the improvement in local accuracy is still remarkable, outperforming FedAvg by 26.3%, 23.6% and 26.9% in the experiments involving 10, 20 and 50 clients, respectively. The local test accuracies of FedHKD* and FedProto are comparable, but FedHKD* outperforms FedProto in terms of global test accuracy (as expected, following the discussion in Section 3.2). FedAlign outperforms the other two regularization methods, FedProx and Moon, both locally and globally; however, but is not competitive with the other methods in which clients' local training is assisted by additional information provided by the server. While it has been reported that FedGen performs well on simpler datasets such as MNIST (LeCun et al., 1998) and EMNIST (Cohen et al., 2017), it appears that its MLP-based gen-

erative model is unable to synthesize data of sufficient quality to assist in KD-based FL on SVHN and CIFAR10/100 – on the former dataset, FedGen actually leads to performance deterioration as compared to FedAvg.

**Training time comparison.** We compare training efficiency of different methods in terms of the averaged training time (in second) per round/client. For fairness, all the experiments were conducted on the same machine with 8 AMD Vega20 GPUs. As shown in Table 1, the training time of FedHKD, FedHKD*, FedProto and FedGen is slightly higher than the training time of FedAvg. The additional computational burden of FedHKD is due to evaluating two extra regularization terms and calculating local hyper-knowledge. The extra computations of FedGen are primarily due to training a generative model; the MLP-based generator leads to minor additional computations but clearly limits the performance of FedGen. FedMD relies on a public dataset of the same size as the clients' local datasets, thus approximately doubling the time FedAvg needs to complete the forward and backward pass during training. Finally, the training efficiency of Moon and FedAlign is inferior to the training efficiency of other methods. Moon is inefficient as it requires more than double the training time of FedAvg. FedAlign needs to pass forward the network multiple times and runs large matrix multiplications to estimate second-order information (Hessian matrix).

**Effect of class heterogeneity.** We compare the performance of the proposed method, FedHKD, and other techniques as the data heterogeneity is varied by tuning the parameter $\beta$. When $\beta = 0.2$, the heterogeneity is severe and the local datasets typically contain only one or two classes; when $\beta = 5$, the local datasets are nearly homogeneous. Data distributions are visualized in Appendix A.2. As shown in Table 2, FedHKD improves both local and global accuracy in all settings, surpassing other methods except FedMD on SVHN dataset for $\beta = 5$. FedProto exhibits remarkable improvement on local accuracy with either extremely heterogeneous ($\beta = 0.2$) or homogeneous ($\beta = 5$) local data but its global performance deteriorates when $\beta = 0.2$.

## 4.3 Privacy Analysis

In our experimental setting, clients share the same network architecture (either ShuffleNetV2 or ResNet18). In both network architectures, the outermost layer in the feature extractor is a batch normalization (BN) layer (Ioffe & Szegedy, 2015). For a batch of vectors $B = \{v_1, \ldots, v_b\}$ at the input of the BN layer, the operation of the BN layer is specified by

$$\mu_B = \frac{1}{b} \sum_{i=1}^{b} v_i, \sigma_B^2 = \frac{1}{b} \sum_{i=1}^{b} (v_i - \mu_B)^2, \tilde{v}_i \leftarrow \frac{v_i - \mu_B}{\sigma_B}. \tag{15}$$

Assuming $b$ is sufficiently large, the law of large numbers implies $\tilde{v}_i \sim \mathcal{N}(0, 1)$. Therefore, $-3 \leq v_i \leq 3$ with probability $99.73\%$ (almost surely). Consider the experimental scenarios where client $i$ contains $N_i = 1024$ samples in its local dataset, the sharing threshold is $\nu = 0.25$, $N_i^j > \nu N_i = 256$, $\delta = 0.01$, and $\epsilon = 0.5$. According to Theorem 1, to obtain 0.5-differential privacy with confidence $1 - \delta = 99\%$ we set $\sigma > \sqrt{2 \log \frac{5}{4\delta}} / \varepsilon \approx 6.215$. According to Lemma 1, $(S_f^i)^2 = \left( \frac{2\zeta}{N_i^j} \right)^2 < (\frac{6}{256})^2$. Setting $\sigma = 7$ (large privacy budget), the variance of noise added to the hyper-knowledge $\mathcal{K}_i^j$ of client $i$ should be $(S_f^i)^2 \sigma^2 < 0.0269$.

## 5 Conclusion

We presented FedHKD, a novel FL algorithm that relies on knowledge distillation to enable efficient learning of personalized and global models in data heterogeneous settings; FedHKD requires neither a public dataset nor a generative model and therefore addresses the data heterogeneity challenge without a need for significantly higher resources. By introducing and utilizing the concept of "hyper-knowledge", information that consists of the means of data representations and the corresponding means of soft predictions, FedHKD enables clients to train personalized models that perform well locally while allowing the server to aggregate a global model that performs well across all data classes. To address privacy concerns, FedHKD deploys a differential privacy mechanism. We conducted extensive experiments in a variety of setting on several benchmark datasets, and provided a theoretical analysis of the convergence of FedHKD. The experimental results demonstrate that FedHKD outperforms state-of-the-art federated learning schemes in terms of both local and global accuracy while only slightly increasing the training time.

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

## A APPENDIX

### A.1 EXPERIMENTAL DETAILS

**General setting.** We implemented all the models and ran the experiments in Pytorch (Paszke et al., 2019) (Ubuntu 18.04 operating system, 8 AMD Vega20 GPUs). Adam (Kingma & Ba, 2014) optimizer was used for model training in all the experiments; learning rate was initialized to 0.001 and decreased every 10 iterations with a decay factor 0.5, while the hyper-parameter $\gamma$ in Adam was set to 0.5. The number of global communication rounds was set to 50 while the number of local epochs was set to 5. The size of a data batch was set to 64 and the participating rate of clients was for simplicity set to 1. For SVHN (Netzer et al., 2011) dataset, the latent dimension of data representation was set to 32; for CIFAR10/100 (Krizhevsky et al., 2009), the latent dimension was set to 64.

**Hyper-parameters.** In all experiments, the FedProx (Li et al., 2020) hyper-parameter $\mu_{prox}$ was set to 0.5; the Moon (Li et al., 2021a) hyper-parameter $\mu_{moon}$ in the proximTal term was set to 1. In FedAlign (Mendieta et al., 2022), the fractional width of the sub-network was set to 0.25, and the balancing parameter $\mu_{align}$ was set to 0.45. The generative model required by FedGen (Zhu et al., 2021) is the MLP-based architecture proposed in (Zhu et al., 2021). The hidden dimension of the generator was set to 512; the latent dimension, noise dimension, and input/output channels were adapted to the datasets. The number of epochs for training the generative model in each global round was set to 5, and the ratio of the generating batch-size and the training batch-size was set to 0.5 (i.e, the generating batch-size was set to 32). Parameters $\alpha_{generative}$ and $\beta_{generative}$ were initialized to 10 with a decay factor 0.98 in each global round. In FedMD (Li & Wang, 2019), we set the regularization hyper-parameter $\lambda_{md}$ to 0.05; the size of the public dataset was set equal to the size of the clients' local training dataset. In FedProto (Tan et al., 2021), the regularization hyper-parameter $\lambda_{proto}$ was set to 0.05. The hyper-parameters $\lambda$ and $\gamma$ in our proposed method FedHKD* were set to 0.05 and 0, respectively; as for FedHKD, the two hyper-parameters $\lambda$ and $\gamma$ were set to 0.05 and 0.05, respectively. Variance $\sigma$ of the Gaussian noise added to the generated hyper-knowledge was set to 7; threshold $\nu$ that needs to be met to initiate computation of hyper-knowledge was set to 0.25. Temperature for FedHKD and Moon algorithm was set to 0.5.

### A.2 DATA PARTITIONING

For convenience, we used datasets encapsulated by Torchvision To obtain the global test dataset, we directly load SVHN, CIFAR10 and CIFAR100 test set in Torchvision without any sampling. For the local training and test sets, we first utilized Dirichlet distribution to sample $m$ partitions as $m$ local datasets from the encapsulated set ($m$ denotes the number of clients). Then we divided the local dataset into a training and test set in 75%/25% proportion. Figures 1, 2 and 3 visualize the class distribution of local clients by showing the number of samples belonging to different classes at each client (colors distinguish the magnitude – the darker the color, the more samples are in the corresponding class).

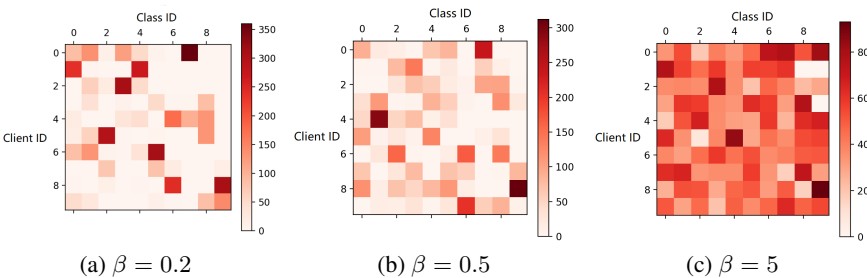

(a) $\beta = 0.2$        (b) $\beta = 0.5$        (c) $\beta = 5$

Figure 1: 10% of the training set points in CIFAR10 are sampled into 10 partitions according to a Dirichlet distribution (10 clients). As the concentration parameter varies ($\beta = 0.2, 0.5, 5$), the partitions change from heterogeneous to homogeneous.

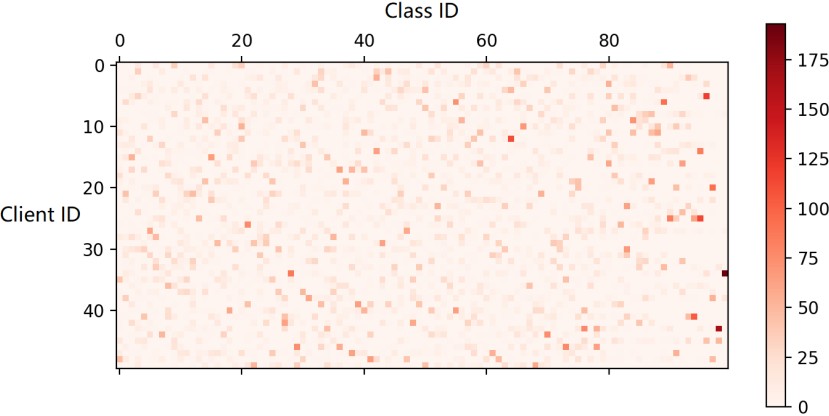

Figure 2: 50% of the training set points in CIFAR10 are sampled into 10 partitions according to a Dirichlet distribution (50 clients). With concentration parameter $\beta = 0.2$, the partition is extremely heterogeneous.

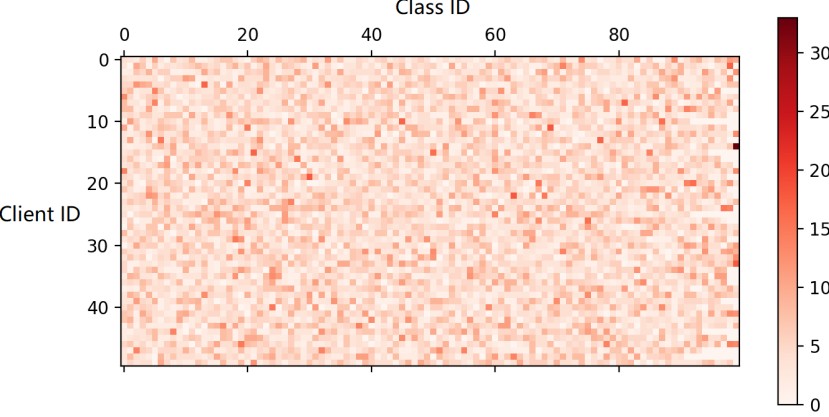

Figure 3: 50% of the training set points in CIFAR100 are sampled into 10 partitions according to a Dirichlet distribution (50 clients). With concentration parameter $\beta = 5$, the partition is relatively homogeneous.

## A.3  FLOW DIAGRAM ILLUSTRATING COMPUTATION OF HYPER-KNOWLEDGE

Figure 4 illustrates computation of local hyper-knowledge by a client. At the end of local training, each participating client obtains a fine-tuned local model consisting of a feature extractor $R_\phi(\cdot)$ and a classifier $G_\omega(\cdot)$. There are three steps in the process of obtaining local hyper-knowledge for class $j$ of client $k$: (1) Representations of data samples in class $j$, generated by the feature extractor, are used to compute the mean of data representations for that class; (2) A classifier generates soft predictions for the obtained data representations, thus enabling computation of the mean of soft predictions for class $j$; (3) After adding Gaussian noise to the mean of data representations, the noisy mean of data representations and mean of soft predictions are packaged into local hyper-knowledge for class $j$.

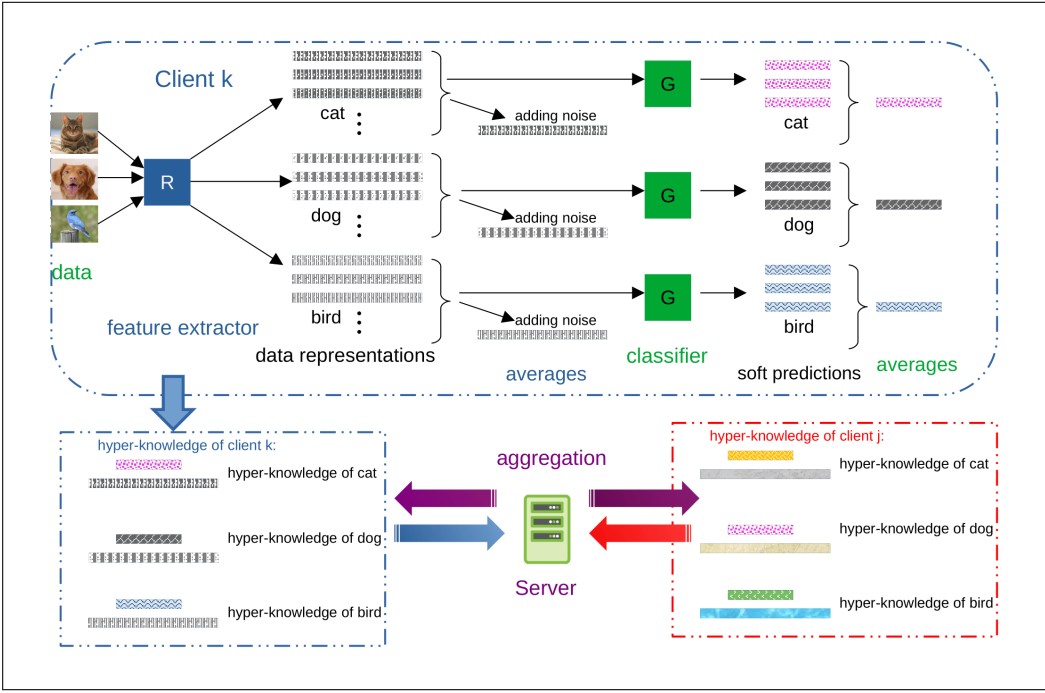

Figure 4: A flow diagram showing computation, encryption and aggregation of hyper-knowledge.

## A.4  DETAILS OF THE FEDHKD ALGORITHM

Figure. 5 illustrates the iterative training procedure of FedHKD. At the start of training, global hyper-knowledge is initialized to an empty set and thus in round 1 each client trains its local model without global hyper-knowledge. Following local training, each client extracts representations from local data samples via a feature extractor and finds soft predictions via a classifier, computing local hyper-knowledge as shown in Figure. 4. The server collects local hyper-knowledge and model updates from clients, aggregates them into global hyper-knowledge and model, and then sends the results back to the clients. From this point on, clients perform local training aided by the global knowledge. Alternating local training and aggregation lasts for $T-1$ rounds where $T$ denotes the number of global epochs.

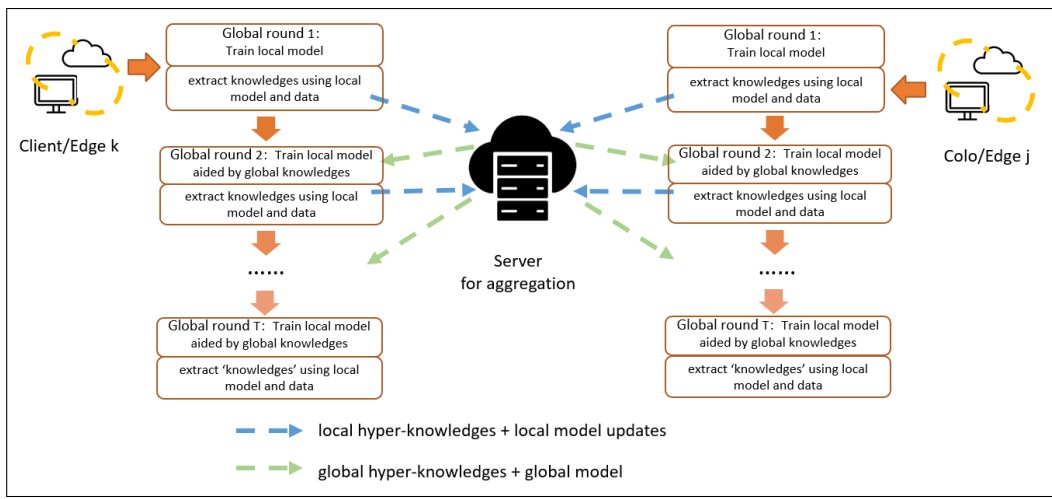

Figure 5: A flow diagram showing FedHKD steps. The blue dashed line indicates sending local hyper-knowledge and model updates from clients to the server while the green dashed line indicates broadcasting global hyper-knowledge and model from the server to clients.

## A.5 Proof of Lemma 1

To compute $i^{th}$ client's mean of class $j$ representation, $\bar{\boldsymbol{h}}_i^j$, we consider the deterministic function (averaging in an element-wise manner) $f_l(\boldsymbol{d}_i^j) \triangleq \bar{\boldsymbol{h}}_i^j(l) = \frac{1}{N_i^j} \sum_{k=1}^{N_i^j} \bar{\boldsymbol{h}}_i^{j,k}(l)$ where $\boldsymbol{d}_i^j$ is the subset of the $i^{th}$ client's local dataset collecting samples with label $j$; $\boldsymbol{h}_i^{j,k}$ denotes the data representation of the $k^{th}$ sample in $\boldsymbol{d}_i^j$ while $\boldsymbol{h}_i^{j,k}(l)$ is the $l^{th}$ element of $\boldsymbol{h}_i^{j,k}$.

**Lemma 1.** If $|\boldsymbol{h}_i^{j,k}(l)|$ is bounded by $\zeta > 0$ for any $k$, then

$$|f_l(\boldsymbol{d}_i^j) - f_l(\boldsymbol{d}_i^{j\prime})| \leq \frac{2\zeta}{N_i^j}. \tag{16}$$

*Proof:* Without a loss of generality, specify

$$\boldsymbol{e} = \{h_i^1(l), \ldots, h_i^{N_i^j - 1}(l), h_i^{N_i^j}(l)\}, \ |\boldsymbol{e}| = N_i^j, \tag{17}$$

and

$$\boldsymbol{e}' = \{h_i^1(l), \ldots, h_i^{N_i^j - 1}(l)\}, \ |\boldsymbol{e}'| = N_i^j - 1, \tag{18}$$

where $\boldsymbol{e}$ and $\boldsymbol{e}'$ denote adjacent sets differing in at most one element. Define $\boldsymbol{1} = \{1, \ldots, 1\}$ with $|\boldsymbol{1}| = N_i^j - 1$. Then

$$
\begin{aligned}
|f_l(\boldsymbol{d}_i^j) - f(\boldsymbol{d}_i^{j'})| &= \left| \frac{\boldsymbol{1}^T \boldsymbol{e}' + h_i^{N_i^j}(l)}{N_i^j} - \frac{\boldsymbol{1}^T \boldsymbol{e}'}{N_i^j - 1} \right| \\
&= \left| \frac{\left( N_i^j - 1 \right) h_i^{N_i^j}(l) - \boldsymbol{1}^T \boldsymbol{e}'}{N_i^j \left( N_i^j - 1 \right)} \right| \\
&\leq \left| \frac{\left( N_i^j - 1 \right) h_i^{N_i^j}(l)}{N_i^j \left( N_i^j - 1 \right)} \right| + \left| \frac{\boldsymbol{1}^T \boldsymbol{e}'}{N_i^j \left( N_i^j - 1 \right)} \right| \\
&\leq \left| \frac{\left( N_i^j - 1 \right) \zeta}{N_i^j \left( N_i^j - 1 \right)} \right| + \left| \frac{\left( N_i^j - 1 \right) \zeta}{N_i^j \left( N_i^j - 1 \right)} \right| \\
&= \frac{\zeta}{N_i^j} + \frac{\zeta}{N_i^j} = \frac{2\zeta}{N_i^j}.
\end{aligned}
\tag{19}
$$

A.6 CONVERGENCE ANALYSIS OF FEDHKD

It will be helpful to recall the notation before restating the theorems and providing their proofs. Let $R_{\phi_i}(\cdot) : \mathbb{R}^{d_x} \to \mathbb{R}^{d_r}$ denote the feature extractor function of client $i$, mapping the raw data of dimension $d_x$ into the representation space of dimension $d_r$. Let $G_{\omega_i}(\cdot) : \mathbb{R}^{d_r} \to \mathbb{R}^n$ denote the classifier's function of client $i$, projecting the data representation into the categorical space of dimension $n$. Let $F_{\theta_i = (\phi_i, \omega_i)}(\cdot) = G_{\omega_i}(\cdot) \circ R_{\phi_i}(\cdot)$ denote the mapping of the entire model. The local objective function of client $i$ is formed as

$$
\begin{aligned}
\mathcal{L}(\mathcal{D}_i, \phi_i, \omega_i) = {} & \frac{1}{B_i} \sum_{k=1}^{B_i} \textbf{CELoss}(G_{\omega_i}(R_{\phi_i}(\boldsymbol{x}_k)), y_k) \\
& + \lambda \frac{1}{n} \sum_{j=1}^{n} \|Q(G_{\omega_i}(\mathcal{H}^j), T) - \mathcal{Q}^j\|_2 + \gamma \frac{1}{B_i} \sum_{k=1}^{B_i} \|R_{\phi_i}(\boldsymbol{x}_k) - \mathcal{H}^{y_k}\|_2,
\end{aligned}
\tag{20}
$$

where $\mathcal{D}_i$ denotes the local dataset of client $i$; input $\boldsymbol{x}_k$ and label $y_k$ are drawn from $\mathcal{D}_i$; $B_i$ is the number of samples in a batch of $\mathcal{D}_i$; $Q(\cdot, T)$ is the soft target function with temperature $T$; $\mathcal{H}^j$ denotes the global mean data representation of class $j$; $\mathcal{Q}^{y_k}$ is the corresponding global soft prediction of class $y_k$; and $\lambda$ and $\gamma$ are the hyper-parameters. Note that only $\phi_i$ and $\omega_i$ are variables in the loss function while the other terms are constant.

Let $t$ denote the current global training round. During any global round, there are $E$ local training epochs. Assume the loss function is minimized by relying on stochastic gradient descent (SGD). To compare the loss before and after model/hyper-knowledge aggregation at the server, denote the local epoch by $e \in \{\frac{1}{2}, 1, \dots, E\}$; $e = \frac{1}{2}$ indicates the epoch between the end of the server's aggregation in the previous communication round and the first epoch of the local training in the next round. After $E$ epochs of local training in communication round $t$, the local model of client $i$ is denoted as $(\phi_i^{E,t}, \omega_i^{E,t})$. At the global communication round $t + 1$, client $i$ initializes the local model with the aggregated global model, $(\phi_i^{\frac{1}{2}, t+1}, \omega_i^{\frac{1}{2}, t+1})$. Although client $i$ does not begin the next training epoch, the local model is changed and so is the output of the loss function. At the server, the global model is updated as

$$
\boldsymbol{\theta}^{\frac{1}{2}, t+1} = \sum_{i=1}^{m} p_i \boldsymbol{\theta}_i^{E, t},
\tag{21}
$$

where $\boldsymbol{\theta}_i^{E,t}$ is the local model of client $i$ after $E$ local training epoches at round $t$; $p_i$ is the averaging weight of client $i$, where $\sum_{i=1}^{m} p_i = 1$. $\tilde{\boldsymbol{h}}^{j,t}$ and $\bar{\boldsymbol{q}}^{j,t}$ are aggregated as

$$
\mathcal{H}^{j, t+1} = \sum_{i=1}^{m} p_i \tilde{\boldsymbol{h}}^{j, t},
\tag{22}
$$

$$
\mathcal{Q}^{j, t+1} = \sum_{i=1}^{m} p_i \bar{\boldsymbol{q}}^{i, t}.
\tag{23}
$$

A.6.1 ASSUMPTIONS

**Assumption 1.** (Lipschitz Continuity). The gradient of the local loss function $\mathcal{L}(\cdot)$ is $L_1$-Lipschitz continuous, the embedding functions of the local feature extractor $R_\phi(\cdot)$ is $L_2$-Lipschitz continuous, and the embedding functions of the local classifier $G_\omega(\cdot)$ composition with soft prediction function $Q(\cdot, T)$ is $L_3$-Lipschitz continuous,

$$
\left\|\nabla\mathcal{L}(\boldsymbol{\theta}^{t_1}) - \nabla\mathcal{L}(\boldsymbol{\theta}^{t_2})\right\|_2 \le L_1 \left\|\boldsymbol{\theta}^{t_1} - \boldsymbol{\theta}^{t_2}\right\|_2, \forall t_1, t_2 > 0,
\tag{24}
$$

$$
\left\|R_{\phi^{t_1}}(\cdot) - R_{\phi^{t_2}}(\cdot)\right\| \le L_2 \left\|\phi^{t_1} - \phi^{t_2}\right\|_2, \quad \forall t_1, t_2 > 0,
\tag{25}
$$

$$
\left\|Q(G_{\omega^{t_1}}(\cdot)) - Q(G_{\omega^{t_2}}(\cdot))\right\| \le L_3 \left\|\omega^{t_1} - \omega^{t_2}\right\|_2, \quad \forall t_1, t_2 > 0.
\tag{26}
$$

Inequality 24 also implies

$$
\mathcal{L}(\boldsymbol{\theta}^{t_1}) - \mathcal{L}(\boldsymbol{\theta}^{t_2}) \le \left\langle \nabla\mathcal{L}(\boldsymbol{\theta}^{t_2}), \boldsymbol{\theta}^{t_1} - \boldsymbol{\theta}^{t_2} \right\rangle + \frac{L_1}{2} \left\|\boldsymbol{\theta}^{t_1} - \boldsymbol{\theta}^{t_2}\right\|_2^2, \quad \forall t_1, t_2 > 0.
\tag{27}
$$

**Assumption 2.** (Unbiased Gradient and Bounded Variance). The stochastic gradients on a batch of client $i$'s data $\xi_i$, denoted by $\boldsymbol{g}_i^t = \nabla\mathcal{L}(\boldsymbol{\theta}_i^t, \xi_i^t)$, is an unbiased estimator of the local gradient for each client $i$,

$$\mathbb{E}_{\xi_i \sim D_i}\left[\boldsymbol{g}_i^t\right] = \nabla\mathcal{L}\left(\boldsymbol{\theta}_i^t\right) \quad \forall i \in 1, 2, \ldots, m, \tag{28}$$

with the variance bounded by $\sigma^2$,

$$\mathbb{E}\left[\left\|\boldsymbol{g}_i^t - \nabla\mathcal{L}\left(\boldsymbol{\theta}_i^t\right)\right\|_2^2\right] \leq \sigma^2, \quad \forall i \in \{1, 2, \ldots, m\},\ \sigma > 0. \tag{29}$$

**Assumption 3.** (Bounded Expectation of Gradients). The expectation of the stochastic gradient is bounded by $V$,

$$\mathbb{E}\left[\left\|\boldsymbol{g}_i^t\right\|_2^2\right] \leq V^2, \quad \forall i \in \{1, 2, \ldots, m\},\ V > 0. \tag{30}$$

### A.6.2 LEMMAS

**Lemma 2.** Instate Assumptions 1-3. The loss function after $E$ local training epoches at global round $t + 1$ can be bounded as

$$\mathbb{E}\left[\mathcal{L}^{E,t+1}\right] \overset{(1)}{\leq} \mathcal{L}^{\frac{1}{2},t+1} - \sum_{e=\frac{1}{2}}^{E-1}\left(\eta_e - \frac{\eta_e^2 L_1}{2}\right)\left\|\nabla\mathcal{L}^{e,t+1}\right\|_2^2 + \frac{\eta_0^2 L_1 E}{2}\sigma^2, \tag{31}$$

where $\eta_e$ is the step-size (learning rate) at local epoch $e$.

*Proof:*

$$\begin{aligned}
\mathcal{L}^{e+1,t+1} &\overset{(1)}{\leq} \mathcal{L}^{e,t+1} + \left\langle\nabla\mathcal{L}^{e,t+1}, \boldsymbol{\theta}^{e+1,t+1} - \boldsymbol{\theta}^{e,t+1}\right\rangle + \frac{L_1}{2}\left\|\boldsymbol{\theta}^{e+1,t+1} - \boldsymbol{\theta}^{e,t+1}\right\|_2^2 \\
&= \mathcal{L}^{e,t+1} - \eta_e\left\langle\nabla\mathcal{L}^{e,t+1}, \boldsymbol{g}^{e,t+1}\right\rangle + \frac{L_1}{2}\eta_e^2\left\|\boldsymbol{g}^{e,t+1}\right\|_2^2, e \in \{\frac{1}{2}, 1, \ldots, E-1\},
\end{aligned} \tag{32}$$

where inequality (1) follows from Assumption 1. Taking expectation of both sides (the sampling batch $\xi^{t+1}$), we obtain

$$\begin{aligned}
\mathbb{E}\left[\mathcal{L}^{e+1,t+1}\right] &\overset{(2)}{\leq} \mathcal{L}^{e,t+1} - \eta_e\left\|\nabla\mathcal{L}^{e,t+1}\right\|_2^2 + \frac{L_1}{2}\eta_e^2\mathbb{E}\left[\left\|\boldsymbol{g}^{e,t+1}\right\|_2^2\right] \\
&\overset{(3)}{=} \mathcal{L}^{e,t+1} - \eta_e\left\|\nabla\mathcal{L}^{e,t+1}\right\|_2^2 + \frac{L_1}{2}\eta_e^2\left(\left\|\nabla\mathcal{L}^{e,t+1}\right\|_2^2 + \mathbb{V}\left[\boldsymbol{g}^{e,t+1}\right]\right) \\
&\overset{(4)}{\leq} \mathcal{L}^{e,t+1} - \left(\eta_e - \frac{\eta_e^2 L_1}{2}\right)\left\|\nabla\mathcal{L}^{e,t+1}\right\|_2^2 + \frac{L_1}{2}\eta_e^2\sigma^2.
\end{aligned} \tag{33}$$

Inequality (2) follows from Assumption 2; (3) follows from $\mathbb{V}[x] = \mathbb{E}\left[x^2\right] - \mathbb{E}[x]^2$, where $x$ is a random variable; (4) holds due to Assumptions 2-3. Let us set the learning step at the start of local training to $\eta_{\frac{1}{2}} = \eta_0$. By telescoping,

$$\mathbb{E}\left[\mathcal{L}^{E,t+1}\right] \leq \mathcal{L}^{\frac{1}{2},t+1} - \sum_{e=\frac{1}{2}}^{E-1}\left(\eta_e - \frac{\eta_e^2 L_1}{2}\right)\left\|\nabla\mathcal{L}^{e,t+1}\right\|_2^2 + \frac{\eta_0^2\sigma^2 L_1 E}{2}. \tag{34}$$

The above inequality holds due to the fact that the learning rate $\eta$ is non-increasing.

**Lemma 2.** Following the model and hyper-knowledge aggregation at the server, the loss function of any client $i$ at global round $t + 1$ can be bounded as

$$\mathbb{E}\left[\mathcal{L}_i^{\frac{1}{2},(t+1)}\right] \leq \mathcal{L}_i^{E,t} + \frac{\eta_0^2 L_1}{2}E^2 V^2 + 2\lambda\eta_0 L_3\left(L_2 + 1\right)EV + 2\gamma\eta_0 L_2 EV. \tag{35}$$

*Proof:*

$$
\begin{aligned}
\mathcal{L}_i^{\frac{1}{2},(t+1)} - \mathcal{L}_i^{E,t} &= \mathcal{L}(\boldsymbol{\theta}_i^{\frac{1}{2},t+1}, \mathcal{K}^{t+1}) - \mathcal{L}(\boldsymbol{\theta}_i^{E,t}, \mathcal{K}^t) \\
&= \mathcal{L}(\boldsymbol{\theta}_i^{\frac{1}{2},t+1}, \mathcal{K}^{t+1}) - \mathcal{L}(\boldsymbol{\theta}_i^{E,t}, \mathcal{K}^{t+1}) + \mathcal{L}(\boldsymbol{\theta}_i^{E,t}, \mathcal{K}^{t+1}) - \mathcal{L}(\boldsymbol{\theta}_i^{E,t}, \mathcal{K}^t) \\
&\overset{(1)}{\leq} \left\langle \nabla\mathcal{L}_i^{E,t}, \boldsymbol{\theta}_i^{\frac{1}{2},t+1} - \boldsymbol{\theta}_i^{E,t} \right\rangle + \frac{L_1}{2} \left\| \boldsymbol{\theta}_i^{\frac{1}{2},t+1} - \boldsymbol{\theta}_i^{E,t} \right\|_2^2 \\
&\quad + \mathcal{L}(\boldsymbol{\theta}_i^{E,t}, \mathcal{K}^{t+1}) - \mathcal{L}(\boldsymbol{\theta}_i^{E,t}, \mathcal{K}^t) \\
&\overset{(2)}{=} \left\langle \nabla\mathcal{L}_i^{E,t}, \sum_{j=1}^m p_j \boldsymbol{\theta}_j^{E,t} - \boldsymbol{\theta}_i^{E,t} \right\rangle + \frac{L_1}{2} \left\| \sum_{j=1}^m p_j \boldsymbol{\theta}_j^{E,t} - \boldsymbol{\theta}_i^{\frac{1}{2},t} \right\|_2^2 \\
&\quad + \mathcal{L}(\boldsymbol{\theta}_i^{E,t}, \mathcal{K}^{t+1}) - \mathcal{L}(\boldsymbol{\theta}_i^{E,t}, \mathcal{K}^t),
\end{aligned}
\tag{36}
$$

where inequality (1) follows from Assumption 1, and (2) is derived from Eq. 21. Taking expectation of both side,

$$
\begin{aligned}
\mathbb{E}\left[\mathcal{L}_i^{\frac{1}{2},(t+1)}\right] - \mathcal{L}_i^{E,t} &\overset{(1)}{\leq} \frac{L_1}{2}\mathbb{E}\left\| \sum_{j=1}^m p_j \boldsymbol{\theta}_j^{E,t} - \boldsymbol{\theta}_i^{E,t} \right\|_2^2 + \mathbb{E}\mathcal{L}(\boldsymbol{\theta}_i^{E,t}, \mathcal{K}^{t+1}) - \mathbb{E}\mathcal{L}(\boldsymbol{\theta}_i^{E,t}, \mathcal{K}^t) \\
&= \frac{L_1}{2}\mathbb{E}\left\| \sum_{j=1}^m p_j \boldsymbol{\theta}_j^{E,t} - \boldsymbol{\theta}_i^{\frac{1}{2},t} - \left( \boldsymbol{\theta}_i^{E,t} - \boldsymbol{\theta}_i^{\frac{1}{2},t} \right) \right\|_2^2 \\
&\quad + \mathbb{E}\mathcal{L}(\boldsymbol{\theta}^{E,t}, \mathcal{K}^{t+1}) - \mathbb{E}\mathcal{L}(\boldsymbol{\theta}^{E,t}, \mathcal{K}^t) \\
&\overset{(2)}{\leq} \frac{L_1}{2}\mathbb{E}\left\| \boldsymbol{\theta}_i^{E,t} - \boldsymbol{\theta}_i^{\frac{1}{2},t} \right\|_2^2 + \mathbb{E}\mathcal{L}(\boldsymbol{\theta}^{E,t}, \mathcal{K}^{t+1}) - \mathbb{E}\mathcal{L}(\boldsymbol{\theta}^{E,t}, \mathcal{K}^t) \\
&= \frac{L_1}{2}\mathbb{E}\left\| \sum_{e=\frac{1}{2}}^{E-1} \eta_e \boldsymbol{g}_i^{e,t} \right\|_2^2 + \mathbb{E}\mathcal{L}(\boldsymbol{\theta}^{E,t}, \mathcal{K}^{t+1}) - \mathbb{E}\mathcal{L}(\boldsymbol{\theta}^{E,t}, \mathcal{K}^t) \\
&\overset{(3)}{\leq} \frac{L_1}{2}\mathbb{E}\sum_{e=\frac{1}{2}}^{E-1} E\eta_e^2 \left\| \boldsymbol{g}_i^{e,t} \right\|_2^2 + \mathbb{E}\mathcal{L}(\boldsymbol{\theta}^{E,t}, \mathcal{K}^{t+1}) - \mathbb{E}\mathcal{L}(\boldsymbol{\theta}^{E,t}, \mathcal{K}^t) \\
&\overset{(4)}{\leq} \frac{\eta_{\frac{1}{2}}^2 L_1}{2}\mathbb{E}\sum_{e=\frac{1}{2}}^{E-1} E \left\| \boldsymbol{g}_i^{e,t} \right\|_2^2 + \mathbb{E}\mathcal{L}(\boldsymbol{\theta}^{E,t}, \mathcal{K}^{t+1}) - \mathbb{E}\mathcal{L}(\boldsymbol{\theta}^{E,t}, \mathcal{K}^t) \\
&\overset{(5)}{\leq} \frac{\eta_0^2 L_1}{2} E^2 V^2 + \mathbb{E}\mathcal{L}(\boldsymbol{\theta}^{E,t}, \mathcal{K}^{t+1}) - \mathbb{E}\mathcal{L}(\boldsymbol{\theta}^{E,t}, \mathcal{K}^t).
\end{aligned}
\tag{37}
$$

Due to Lemma 3 and the proof of Lemma 3 in (Li et al., 2019), inequality (1) holds as $\mathbb{E}\left[\boldsymbol{\theta}_j^{E,t}\right] = \sum_{j=1}^m p_j \boldsymbol{\theta}_j^{E,t}$; inequality (2) holds because $\mathbb{E}\|\mathbb{E}X - X\|^2 \leq \mathbb{E}\|X\|^2$, where $X = \boldsymbol{\theta}_i^{E,t} - \boldsymbol{\theta}_i^{\frac{1}{2},t}$; inequality (3) is due to Jensen inequality; inequality (4) follows from that fact that the learning rate $\eta_e$ is non-increasing; inequality (5) holds due to Assumption 3. Let us consider the term $\mathcal{L}(\boldsymbol{\theta}^{E,t}, \mathcal{K}^{t+1}) - \mathcal{L}(\boldsymbol{\theta}^{E,t}, \mathcal{K}^t)$; note that the model parameters $\boldsymbol{\theta}^{E,t}$ are unchanged and thus the first term in the loss function 20 can be neglected. The difference between the two loss functions is

due to different global hyper-knowledge $\mathcal{K}^t$ and $\mathcal{K}^{t+1}$, $\mathcal{L}(\boldsymbol{\theta}^{E,t}, \mathcal{K}^{t+1}) - \mathcal{L}(\boldsymbol{\theta}^{E,t}, \mathcal{K}^t) =$

$$
\begin{aligned}
&= \lambda \frac{1}{n} \sum_{j=1}^{n} \left( \left\| Q\left( G_{\boldsymbol{\omega}_j^{E,t}}(\mathcal{H}^{j,t+1}) \right) - \mathcal{Q}^{j,t+1} \right\|_2 - \left\| Q\left( G_{\boldsymbol{\omega}_j^{E,t}}(\mathcal{H}^{j,t}) \right) - \mathcal{Q}^{j,t} \right\|_2 \right) \\
&\quad + \gamma \frac{1}{B_i} \sum_{k=1}^{B_i} \left( \left\| R_{\boldsymbol{\omega}_i^{E,t}}(\boldsymbol{x}_k) - \mathcal{H}^{y_k,t+1} \right\|_2 - \left\| R_{\boldsymbol{\omega}_i^{E,t}}(\boldsymbol{x}_k) - \mathcal{H}^{y_k,t} \right\|_2 \right) \\
&= \lambda \frac{1}{n} \sum_{j=1}^{n} \left( \left\| Q\left( G_{\boldsymbol{\omega}_j^{E,t}}(\mathcal{H}^{j,t+1}) \right) - \mathcal{Q}^{j,t} + \mathcal{Q}^{j,t} - \mathcal{Q}^{j,t+1} \right\|_2 - \left\| Q\left( G_{\boldsymbol{\omega}_j^{E,t}}(\mathcal{H}^{j,t}) \right) - \mathcal{Q}^{j,t} \right\|_2 \right) \\
&\quad + \gamma \frac{1}{B_i} \sum_{k=1}^{B_i} \left( \left\| R_{\boldsymbol{\omega}_i^{E,t}}(\boldsymbol{x}_k) - \mathcal{H}^{y_k,t+1} \right\|_2 - \left\| R_{\boldsymbol{\omega}_i^{E,t}}(\boldsymbol{x}_k) - \mathcal{H}^{y_k,t} \right\|_2 \right) \\
&\overset{(1)}{\leq} \lambda \frac{1}{n} \sum_{j=1}^{n} \left( \left\| Q\left( G_{\boldsymbol{\omega}_j^{E,t}}(\mathcal{H}^{j,t+1}) \right) - Q\left( G_{\boldsymbol{\omega}_j^{E,t}}(\mathcal{H}^{j,t}) \right) \right\|_2 + \left\| \mathcal{Q}^{j,t+1} - \mathcal{Q}^{j,t} \right\|_2 \right) \\
&\quad + \gamma \frac{1}{B_i} \sum_{k=1}^{B_i} \left( \left\| \mathcal{H}^{y_k,t+1} - \mathcal{H}^{y_k,t} \right\|_2 \right) \\
&\overset{(2)}{\leq} \lambda \frac{1}{n} \sum_{j=1}^{n} \left( L_3 \left\| \mathcal{H}^{j,t+1} - \mathcal{H}^{j,t} \right\|_2 + \left\| \mathcal{Q}^{j,t+1} - \mathcal{Q}^{j,t} \right\|_2 \right) + \gamma \frac{1}{B_i} \sum_{k=1}^{B_i} \left( \left\| \mathcal{H}^{y_k,t+1} - \mathcal{H}^{y_k,t} \right\|_2 \right),
\end{aligned}
$$
(38)

where (1) is due to the triangle inequality, $\|a + b + c\|_2 \leq \|a\|_2 + \|b\|_2 + \|c\|_2$ with $a = Q\left( G_{\boldsymbol{\omega}_j^{E,t}}(\mathcal{H}^{j,t}) \right) - \mathcal{Q}^{j,t}$, $b = Q\left( G_{\boldsymbol{\omega}_j^{E,t}}(\mathcal{H}^{j,t+1}) \right) - Q\left( G_{\boldsymbol{\omega}_j^{E,t}}(\mathcal{H}^{j,t}) \right)$ and $c = \mathcal{Q}^{j,t} - \mathcal{Q}^{j,t+1}$; inequality (2) holds due to Assumption 1. Then, let us consider the following difference:

$$
\begin{aligned}
\left\| \mathcal{H}^{j,t+1} - \mathcal{H}^{j,t} \right\|_2 &= \left\| \sum_{i=1}^{m} p_i \bar{\boldsymbol{h}}_i^{j,t} - \sum_{i=1}^{m} p_i \bar{\boldsymbol{h}}_i^{j,t-1} \right\|_2 \\
&= \left\| \sum_{i=1}^{m} p_i \left( \bar{\boldsymbol{h}}_i^{j,t} - \bar{\boldsymbol{h}}_i^{j,t-1} \right) \right\|_2 \\
&= \left\| \sum_{i=1}^{m} p_i \left( \frac{1}{N_i^j} \sum_{k=1}^{N_i^j} R_{\boldsymbol{\phi}_i^{E,t}}(\boldsymbol{x}_k) - R_{\boldsymbol{\phi}_i^{E,t-1}}(\boldsymbol{x}_k) \right) \right\|_2 \\
&\overset{(1)}{\leq} \sum_{i=1}^{m} p_i \frac{1}{N_i^j} \sum_{k=1}^{N_i^j} \left\| R_{\boldsymbol{\phi}_i^{E,t}}(\boldsymbol{x}_k) - R_{\boldsymbol{\phi}_i^{E,t-1}}(\boldsymbol{x}_k) \right\|_2 \\
&\overset{(2)}{\leq} \sum_{i=1}^{m} p_i \frac{1}{N_i^j} \sum_{k=1}^{N_i} L_2 \left\| \boldsymbol{\phi}_i^{E,t} - \boldsymbol{\phi}_i^{E,t-1} \right\|_2 \\
&= L_2 \sum_{i=1}^{m} p_i \left\| \boldsymbol{\phi}_i^{E,t} - \boldsymbol{\phi}_i^{E,t-1} \right\|_2 .
\end{aligned}
$$
(39)

Inequality (1) holds due to Jensen's inequality, while inequality (2) follows from Assumption 1.

For convenience (and perhaps clarity), we drop the superscript $j$ denoting the class. Taking expectation of both sides,

$$
\begin{aligned}
\mathbb{E}\left\|\mathcal{H}^{t+1} - \mathcal{H}^t\right\|_2 &\le L_2 \sum_{i=1}^m p_i \mathbb{E}\left\|\phi_i^{E,t} - \phi_i^{E,t-1}\right\|_2 \\
&\overset{(1)}{\le} L_2 \sum_{i=1}^m p_i \left(\mathbb{E}\left\|\phi_i^{E,t} - \phi_i^{\frac{1}{2},t}\right\|_2 + \mathbb{E}\left\|\phi_i^{\frac{1}{2},t} - \phi_i^{E,t-1}\right\|_2\right) \\
&\overset{(2)}{\le} L_2 \sum_{i=1}^m p_i \left(\eta_0 E V + \mathbb{E}\left\|\sum_j p_j \phi_i^{E,t-1} - \phi_i^{E,t-1}\right\|_2\right) \\
&= L_2 \sum_{i=1}^m p_i \left(\eta_0 E V + \mathbb{E}\left\|\sum_j p_j \phi_i^{E,t-1} - \phi_i^{\frac{1}{2},t-1} + \phi_i^{\frac{1}{2},t-1} - \phi_i^{E,t-1}\right\|_2\right) \\
&\overset{(3)}{\le} L_2 \sum_{i=1}^m p_i \left(\eta_0 E V + \sqrt{\mathbb{E}\left\|\sum_j p_j \phi_i^{E,t-1} - \phi_i^{\frac{1}{2},t-1} + \phi_i^{\frac{1}{2},t-1} - \phi_i^{E,t-1}\right\|_2^2}\right) \\
&\overset{(4)}{\le} L_2 \sum_{i=1}^m p_i \left(\eta_0 E V + \sqrt{\mathbb{E}\left\|\phi_i^{\frac{1}{2},t-1} - \phi_i^{E,t-1}\right\|_2^2}\right) \\
&= L_2 \sum_{i=1}^m p_i \left(\eta_0 E V + \sqrt{\mathbb{E}\left\|\sum_{e=\frac{1}{2}}^{E-1} \eta_e g_i^{e,t-1}\right\|_2^2}\right) \\
&\overset{(5)}{\le} L_2 \sum_{i=1}^m p_i \left(\eta_0 E V + \eta_0 E V\right) \\
&= 2\eta_0 L_2 E V,
\end{aligned}
$$
(40)

where (1) follows from the triangle inequality; inequality (2) holds due to Assumption 3 and the update rule of SGD; since $f(x) = \sqrt{x}$ is concave, (3) follows from Jensen's inequality; inequality (4) holds due to the fact that $\mathbb{E}\left\|\mathbb{E}X - X\right\|^2 \le \mathbb{E}\left\|X\right\|^2$, where $X = \phi_i^{E,t-1} - \phi_i^{\frac{1}{2},t-1}$; inequality (5) follows by using the fact that the learning rate $\eta_e$ is non-increasing.

Similarly,

$$
\begin{aligned}
\mathbb{E}\left\|\mathcal{Q}^{t+1} - \mathcal{Q}^t\right\|_2 &\le L_3 \sum_{i=1}^m p_i \mathbb{E}\left\|\omega_i^{E,t} - \omega_i^{E,t-1}\right\|_2 \\
&\le 2\eta_0 L_3 E V
\end{aligned}
$$
(41)

Combining the above inequalities, we have

$$
\mathbb{E}\left[\mathcal{L}_i^{\frac{1}{2},(t+1)}\right] \le \mathcal{L}_i^{E,t} + \frac{\eta_0^2 L_1}{2} E^2 V^2 + 2\lambda \eta_0 L_3 \left(L_2 + 1\right) E V + 2\gamma \eta_0 L_2 E V.
$$
(42)

### A.6.3 THEOREMS

**Theorem 2.** Instate Assumptions 1-3. For an arbitrary client, after each communication round the loss function is bounded as

$$
\begin{aligned}
\mathbb{E}\left[\mathcal{L}_i^{\frac{1}{2},t+1}\right] \le \mathcal{L}_i^{\frac{1}{2},t} &- \sum_{e=\frac{1}{2}}^{E-1} \left(\eta_e - \frac{\eta_e^2 L_1}{2}\right) \left\|\nabla \mathcal{L}^{e,t}\right\|_2^2 + \frac{\eta_0^2 L_1 E}{2} \left(E V^2 + \sigma^2\right) \\
&+ 2\lambda \eta_0 L_3 \left(L_2 + 1\right) E V + 2\gamma \eta_0 L_2 E V.
\end{aligned}
$$
(43)

Fine-tuning the learning rates $\eta_0$, $\lambda$ and $\gamma$ ensures that

$$\frac{\eta_0^2 L_1 E}{2} \left(EV^2 + \sigma^2\right) + 2\lambda\eta_0 L_3 \left(L_2 + 1\right) EV + 2\gamma\eta_0 L_2 EV - \sum_{e=\frac{1}{2}}^{E-1} \left(\eta_e - \frac{\eta_e^2 L_1}{2}\right) \left\|\nabla\mathcal{L}^{e,t}\right\|_2^2 < 0.$$

(44)

**Corollary 1.** (FedHKD convergence) Let $\eta_0 > \eta_e > \alpha\eta_0$ for $e \in \{1, \ldots, E-1\}, 0 < \alpha < 1$. The loss function of an arbitrary client monotonously decreases in each communication round if

$$\alpha\eta_0 < \eta_e < \frac{2\alpha^2 \left\|\nabla\mathcal{L}^{e,t}\right\| - 4\alpha\lambda L_3(L_2+1)V - 4\alpha\gamma L_2 V}{L_1 \left(\alpha^2 \left\|\nabla\mathcal{L}^{e,t}\right\|_2^2 + 1\right) \left(EV^2 + \sigma^2\right)}, \forall e \in \{1, \ldots, E-1\},$$

(45)

where $\alpha$ denotes the hyper-parameter controlling learning rate decay.

*Proof:*
Since $\eta_0 < \frac{\eta_e}{\alpha}$, in each local epoch $e$ we have

$$\frac{\eta_e^2 L_1}{2\alpha^2} \left(EV^2 + \sigma^2\right) + 2\lambda\frac{\eta_e}{\alpha} L_3 \left(L_2 + 1\right) V + 2\gamma\frac{\eta_e}{\alpha} L_2 V - \left(\eta_e - \frac{\eta_e^2 L_1}{2}\right) \left\|\nabla\mathcal{L}^{e,t}\right\|_2^2 < 0. \quad (46)$$

Dividing both sides by $\eta_e$,

$$\frac{\eta_e L_1}{2\alpha^2} \left(EV^2 + \sigma^2\right) + 2\lambda\frac{1}{\alpha} L_3 \left(L_2 + 1\right) V + 2\gamma\frac{1}{\alpha} L_2 V - \left(1 - \frac{\eta_e L_1}{2}\right) \left\|\nabla\mathcal{L}^{e,t}\right\|_2^2 < 0. \quad (47)$$

Factoring out $\eta_e$ on the left hand side yields

$$\left(\frac{L_1}{2\alpha^2} \left(EV^2 + \sigma^2\right) + \frac{L_1}{2} \left\|\nabla\mathcal{L}^{e,t}\right\|_2^2\right) \eta_e < \left\|\nabla\mathcal{L}^{e,t}\right\|_2^2 - 2\lambda\frac{1}{\alpha} L_3 \left(L_2 + 1\right) V - 2\gamma\frac{1}{\alpha} L_2 V. \quad (48)$$

Dividing both sides by $\left(\frac{L_1}{2\alpha^2} \left(EV^2 + \sigma^2\right) + \frac{L_1}{2} \left\|\nabla\mathcal{L}^{e,t}\right\|_2^2\right)$ results in

$$\eta_e < \frac{2\alpha^2 \left\|\nabla\mathcal{L}^{e,t}\right\| - 4\alpha\lambda L_3(L_2+1)V - 4\alpha\gamma L_2 V}{L_1 \left(\alpha^2 \left\|\nabla\mathcal{L}^{e,t}\right\|_2^2 + 1\right) \left(EV^2 + \sigma^2\right)}, \forall e \in \{1, \ldots, E-1\}.$$

(49)

**Theorem 3.** (FedHKD convergence rate) Instate Assumptions 1-3 and define regret $\Delta = \mathcal{L}^{\frac{1}{2},1} - \mathcal{L}^*$. If the learning rate is set to $\eta$, for an arbitrary client after

$$T = \frac{2\Delta}{\epsilon E \left(2\eta - \eta^2 L_1\right) - \eta^2 L_1 E \left(EV^2 + \sigma^2\right) - 4\lambda\eta L_3 \left(L_2 + 1\right) EV - 4\gamma\eta L_2 EV}$$

(50)

global rounds ($\epsilon > 0$), it holds that

$$\frac{1}{TE} \sum_{t=1}^{T} \sum_{e=\frac{1}{2}}^{E-1} \left\|\nabla\mathcal{L}^{e,t}\right\|_2^2 \leq \epsilon.$$

(51)

*Proof:*
According to Theorem 1,

$$\frac{1}{TE} \sum_{t=1}^{T} \sum_{e=\frac{1}{2}}^{E-1} \left(\eta - \frac{\eta^2 L_1}{2}\right) \left\|\nabla\mathcal{L}^{e,t}\right\|_2^2 \leq \frac{1}{TE} \sum_{t=1}^{T} \mathcal{L}_i^{\frac{1}{2},t} - \frac{1}{TE} \sum_{t=1}^{T} \mathbb{E}\left[\mathcal{L}_i^{\frac{1}{2},t+1}\right] + \frac{\eta^2 L_1}{2} \left(EV^2 + \sigma^2\right)$$

$$+ 2\lambda\eta L_3 \left(L_2 + 1\right) V + 2\gamma\eta L_2 V$$

$$\leq \frac{1}{TE}\Delta + \frac{\eta^2 L_1}{2} \left(EV^2 + \sigma^2\right) + 2\lambda\eta L_3 \left(L_2 + 1\right) V + 2\gamma\eta L_2 V$$

$$< \epsilon\left(\eta - \frac{\eta^2 L_1}{2}\right).$$

(52)

Therefore,

$$\frac{\Delta}{T} \leq \epsilon E \left(\eta - \frac{\eta^2 L_1}{2}\right) - \frac{\eta^2 L_1 E}{2} \left(EV^2 + \sigma^2\right) - 2\lambda\eta L_3 \left(L_2 + 1\right) EV - 2\gamma\eta L_2 EV, \quad (53)$$

which is equivalent to

$$T \geq \frac{2\Delta}{\epsilon E \left(2\eta - \eta^2 L_1\right) - \eta^2 L_1 E \left(EV^2 + \sigma^2\right) - 4\lambda\eta L_3 \left(L_2 + 1\right) EV - 4\gamma\eta L_2 EV}. \quad (54)$$

