# OpenReview forum: "The Best of Both Worlds: Accurate Global and Personalized Models through Federated Learning with Data-Free Hyper-Knowledge Distillation"
_ICLR.cc/2023/Conference — ICLR 2023 poster_

### Official Review · Reviewer_y1Yi · 2022-10-16

**Confidence:** 3
**Correctness:** 3
**Technical Novelty And Significance:** 3
**Empirical Novelty And Significance:** 2
**Recommendation:** 6

**Clarity, Quality, Novelty And Reproducibility:**

Clarity: The clarity of this work is good. I can follow the paper without much effort.

Novelty: The novelty of this work is good. I appreciate the attempts to use KD without public datasets or generative model.

Quality: I have some doubts on the experimental results of this paper. Some justifications are needed to improve its technical quality.

**Strength And Weaknesses:**

Strengths:
- The proposed FedHKD seems efficient and general. FedHKD does not require local datasets, and does not require training generative models, which makes it more applicable in practice.
- The paper is clearly organized and easy to follow. The main advantages and motivations of FedHKD are clearly stated. The method is clearly described.

Weaknesses and Questions.
- Missing an important related work. I appreciate the authors' attempts to bridge local and global federated learning. However, as the authors study class distribution heterogeneity only, I think FedROD (Chen and Chao, 2022) should be discussed and compared, as FedROD also aims to bridge local and global federated learning under class distribution heterogeneity.
- Experiment results do not comply with existing works. For example, with ResNet18 on CIFAR-10, with Dirichlet label distribution $\beta=0.5$ and 20 clients, FedHKD achieve global accuracy of 0.5735. However, as reported in Fed-ROD, with a simple CNN and $\beta=0.3$ (larger distribution heterogeneity), 20 clients, Fed-ROD achieves 0.768 (even FedAvg achieves 0.686). These numbers differ significantly, and the authors should justify the difference.
- "Local Acc" is not clearly described. First, the evaluated model is not clearly stated (I assume that the models evaluated are the local models before aggregation). Second, it has been studied that FedAvg+local fine-tuning (Cheng et al. 2021) is a powerful baseline in personalized FL. I think is is more appropriate and fair to evaluate local accuracy after some local fine-tuning.
- DP on hyper-knowledge seems not necessary. As aggregating the hyper-knowledge requires only addition, is it possible to leverage secure aggregation (Bonawitz et al. 2017) instead of DP, which adds no noise?
- Minor points: It is suggested to replace \cite with \citep in some places, such as Section 4.1. Also, in conclusion, there is a 'FedHDK' which should be a typo.

H. Chen and W. Chao, ON BRIDGING GENERIC AND PERSONALIZED FEDERATED LEARNING FOR IMAGE CLASSIFICATION. ICLR 2022

K. Bonawitz et al. Practical Secure Aggregation for Privacy-Preserving Machine Learning, CCS 2017

Gary Cheng et al. Federated Asymptotics: a model to compare federated learning algorithms. https://arxiv.org/abs/2108.07313


**Summary Of The Paper:**

This paper studies the problem of global and personalized federated learning, aiming to improve both at the same time. This paper proposes a method, FedHKD, that leverages hyper-knowledge distillation to improve both global and local learning. Specifically, hyper-knowledge is defined as the class-wise mean representations and mean logit predictions. The server aggregates meta-knowledge from all clients, and sends the aggregated hyper-knowledge back for clients to learn. Compared with other works which use knowledge distillation to improve FL, FedHKD ahas the advantage of not requiring public datasets, and does not require training generative models. The authors present theoretical analysis on the convergence of FedHKD. Empirically, FedHKD outperforms state-of-the-art federated learning methods in both local and global accuracy.

**Summary Of The Review:**

My main concerns of this paper are 1) missing a related work, and 2) experimental results require more justification. I recommend a weak reject at this stage. I am open to recommending accept if the authors can clarify my doubts.

---

### Official Review · Reviewer_GssC · 2022-10-24

**Confidence:** 4
**Correctness:** 3
**Technical Novelty And Significance:** 3
**Empirical Novelty And Significance:** 3
**Recommendation:** 6

**Clarity, Quality, Novelty And Reproducibility:**

This paper is easy to follow. The quality, clarity and originality are good.



**Strength And Weaknesses:**

Strength:
1. The authors do a lot of experiments to prove that their method is good and compare with many existing methods.
2. The related works are sufficient.
3. The authors provide rigorous theoretical analysis, including privacy analysis and conversion analysis.

Weaknesses:

1. The proposed framework is similar to the paper Fair and Robust Federated Learning Through Personalization.
They all combine data replication to calculate the global model and local model. In the reviewer's opinion,
there is a slight lack of novelty.
2. It will be clearer if there is a structure flow diagram to explain how the proposed scheme works.
3. The author does not explain why the loss function is constructed as in the paper. In other words,
why can't the regularizer be in other forms.



**Summary Of The Paper:**

In this paper the authors propose FedHKD (Federated Hyper-Knowledge Distillation), a novel FL framework
that relies on prototype learning and knowledge distillation to facilitate training on heterogeneous
data. The clients in FedHKD compute mean representations and the corresponding mean soft predictions
(they are called "hyper-knowledge" in the paper) for the data classes in their local training sets. And then,
the hyper-knowledge. is endued by differential privacy via the Gaussian mechanism and sent for aggregation
to the server. The resulting globally aggregated hyper-knowledge is used by clients in the subsequent training
epoch and helps lead to better personalized and global performance.
The main contributions of this paper is:
1. The authors propose a framework called FedHKD, which trains both the global model and the local models well
in federated settings.
2. provide a detailed privacy analysis and a convergence analysis.
3. make extensive experiments to prove FedHKD works.



**Summary Of The Review:**

This paper is good in general. It would be better if the author explain the origin of the loss function.

---

### Official Review · Reviewer_HZm8 · 2022-10-27

**Confidence:** 3
**Correctness:** 4
**Technical Novelty And Significance:** 3
**Empirical Novelty And Significance:** Not applicable
**Recommendation:** 8

**Clarity, Quality, Novelty And Reproducibility:**

The paper is clearly presented, but due to lack of explanation is hard to follow in places.
The ideas seem novel.
There seems to be a lack of details to allow for full reproduction.

**Strength And Weaknesses:**

Strengths:
1. The paper is technically well presented.
2. The idea appears novel and sound.

Weaknesses:
1. A number of the theorems in the main body of the paper do not appear to be used within the main body. It would seem to make more sense to put these into the supplementary material so that more space can be used for better presentation of the main ideas.
2. A number of equations are presented without clear explanation of what they mean.

**Summary Of The Paper:**

The paper presents an extension to the idea of federated learning which works in the situation of heterogeneous data. This is achieved without the need for public data. The approach provides security through the use of differential privacy for the exchanged data.

**Summary Of The Review:**

The paper reads well, but a lack of details on what is going on makes it hard to follow in places. Moving some of the theorems to supplementary material would give more space for further explanation.

---

### Official Review · Reviewer_yjPN · 2022-10-29

**Confidence:** 3
**Correctness:** 3
**Technical Novelty And Significance:** 2
**Empirical Novelty And Significance:** 2
**Recommendation:** 6

**Clarity, Quality, Novelty And Reproducibility:**

The core contribution appears to be making a system that, relative to FedProto, has three changes.  Clients send soft predictions (in addition to class prototypes), the server maintains a central model and sends weights to clients who have the same model architecture, and the client's loss function includes a regularization term to incorporate the soft predictions.

Though the technical contribution is relatively moderate given the above, the work still presents a valuable point in the design space for FL researchers to explore.

Overall, the technical writing in the paper (Section 3) was organized and clear.   However, it was hard to place the work relative to other systems, mostly because it was difficult to determine the exact problem the work addresses.  The paper wants to establish a "best of both worlds", but that thesis doesn't appear well articulated.  While systems like FedProto and KD admit models that are personalized in the sense of statistical distribution and model architecture, FedHKD only supports the first.

While we understand that the technical design borrows from prototypes and KD, its behavior doesn't seem to "achieve the best of both worlds."   In the performance section, FedProto has a global model but in its own words "the global model is a set of class prototypes."  While one issue is that we're not told how you build that global model, the outcome is that FedProto appears to already create good global and local models.

Unfortunately the performance of the system isn't always dominating over the other systems, and the lessons learned from the experiments are hard to tease out.

Other items:
* In general FL papers don't seem to have much empirical evidence for real-world statistical heterogeneity.   Is b=0.5 the right point to evaluate?  I wish the "Effect of class heterogeneity" paragraph had come much earlier in the evaluation section to discuss that parameter.
* What's the impact of the parameter v in terms of performance?
* The addition of differential privacy appears to be a bit of tack-on -- interesting/cool but not fundamental to the techniques presented.
* Perhaps it would be useful to demonstrate the problem in the beginning of the paper, i.e., show where current approaches don't perform well and why/when it actually happens.
* In the sense that FedHKD doesn't send client gradients to the server, it does reduce communication overheads (but not as much as FedProto).  Is that something that matters?
* It isn't clear if local model accuracy is computed with local test sets or global ones.   Sec 4.1, "Datasets" paragraph states that the local models are evaluated on their local data; perhaps explicitly state this is the case for the evaluation numbers.
* It would seem useful to have a column in Table 1 to have a local "best" baseline, and a global "best" baseline.
* While FedHKD* removes the class prototypes, but what about the contribution of sending shared central model weights?


**Strength And Weaknesses:**

Strengths
* Presents an interesting point in the design space between recent work in prototype and KD-based FL systems
* Evaluated on different a handful of different models and datasets, while exploring different client data distributions
* Provides theoretical convergence analysis of the FL system

Weaknesses
* The paper borrows from prior work (which is fine), but it doesn't clearly articulate how it's the "best of both worlds."
* There isn't a clear set of objectives for the work.  Is it to build good personalized or local models?  Is it to build the best global model in the presence of non-iid data?  What is the real-world scenario where the problem your addressing rises?
* The evaluation lacks clear lessons learned.  The system is competitive but not dominating.
* It is not clear what the benefits are of the resulting global model vs. the local models.  What about testing a version of FLHKD where the global model isn't sent to the clients?  What was the penalty in terms of bandwidth relative to FedProto?

**Summary Of The Paper:**

This work presents a federated learning technique to produce good local models and a good global model.  They do so by designing a system that trains a global model without communicating client gradients to the server.  Their system extends recent FL work, which trains local models with client mean class prototypes (FedProto), by also sending client mean (soft) predictions and training a global model.  They call this "Hyper Knowledge." Thus FedHKD is a hybrid approach:  like pure prototype or knowledge distillation, clients avoid sending gradients or parameters, but clients still receive weights from a global model to update their local models.   They evaluate the approach against a number of competing FL systems across three data sets and different levels of non-iid client training data distributions.  Their approach produces models that often do as well and sometimes better than other systems.

**Summary Of The Review:**

The paper presents a useful design point in the very large space of FL systems.  However the paper doesn't make the system objective clear or provide anecdotal or empirical evidence for the target operating regime.  While it borrows from personalized FL and KD FL systems, it doesn't provide for model architecture heterogeneity.   The initial results appear promising but not necessarily dominating over competing systems in the experiments shown.

[Post Author Response]
After reading through the discussion, the author's response has clarified issues (e.g., pointing out the significance of the DiffP piece), and though I remain on the fence about this work, I'm comfortable adjusting the score to a 6 (marginally above).   There is no doubt the work makes some contributions in the FL space that builds personalized and global models.   FedHKD builds mostly better local and global models when comparing to a scheme that is also data-free and admits different client network architectures.   But if one says the performance results are solid but not dominating over another data-free scheme, the claim that being data-free is a significant contribution seems to be weakened.

---

### Decision · Program_Chairs · 2023-01-20

**Decision:**

Accept: poster

**Justification For Why Not Higher Score:**

The theorems in the paper do not directly give insights into the final algorithm. The DP theorem seems quite standard with Gaussian mechanism.  The ideas of knowledge distillation is also not quite well formed but results in an interesting approach.

**Justification For Why Not Lower Score:**

The paper did a very good job of mathematically formalizing their arguments. Experiments are quite rigorous with extensive set of baselines. The paper will be a good read.

**Metareview: Summary, Strengths And Weaknesses:**

The paper proposes a fundamental framework FedHKD to address the problem of imbalanced class local data distribution.   FedHKD used Knoelwdge distillation to trains both the global model and the local models well in federated settings.  The paper provide a detailed privacy analysis and a convergence analysis.  A very thorough and extensive experiments to demonstrate the need for the proposed FedHKD.

The paper provides a very convincing and formal arguments which is rare in most Federated learning papers. The experiments are also very rigorous and convincing.



**Note From Pc:**

if the above contains the word "oral" or "spotlight" please see: "oral" presentation means -> notable-top-5% and "spotlight" means -> notable-top-25%. As stated in our emails, we are disassociating presentation type from AC recommendations

**Summary Of Ac-Reviewer Meeting:**


Reviewers unanimously agreed that the contribution of the paper is worthy of publication.